# Assessing transient changes in the ocean carbon cycle during the last deglaciation through carbon isotope modeling

Hidetaka Kobayashi[1,2], Akira Oka[2], Takashi Obase[2], and Ayako Abe-Ouchi[2]

[1]Faculty of Science, Academic Assembly, University of Toyama, 3190 Gofuku, Toyama, 930-8555, Japan.
[2]Atmosphere and Ocean Research Institute, University of Tokyo, 5-1-5 Kashiwanoha, Kashiwa, 277-8568, Japan.

**Correspondence:** H. Kobayashi (hidekoba@sci.u-toyama.ac.jp)

**Abstract.**

Atmospheric carbon dioxide concentration ($pCO_2$) has increased by approximately $80$ ppm from the Last Glacial Maximum (LGM) to the early Holocene. The change in this atmospheric greenhouse gas is recognized as a climate system response to gradual change in insolation. Previous modeling studies suggested that the deglacial increase in atmospheric $pCO_2$ is primarily attributed to the release of $CO_2$ from the ocean. Additionally, it has been suggested that abrupt change in the Atlantic Meridional Overturning Circulation (AMOC) and associated interhemispheric climate changes are involved in the release of $CO_2$. However, understanding remains limited regarding oceanic circulation changes, and the factors responsible for changes in chemical tracers in the ocean during the last deglaciation and their impact on atmospheric $pCO_2$. In this study, we investigated the evolution of the ocean carbon cycle during the last deglaciation (21 to 11 ka BP) using three-dimensional ocean fields from the transient simulation of the MIROC 4m climate model, which exhibits abrupt AMOC changes similar to those observed in reconstructions. We investigate the reliability of simulated changes in the ocean carbon cycle by comparing the simulated carbon isotope ratios with sediment core data, and examine potential biases and overlooked or underestimated processes in the model. Qualitatively, the modeled changes in atmospheric $pCO_2$ are consistent with ice core records. For example, during Heinrich Stadial 1 (HS1), atmospheric $pCO_2$ increases by $10.2$ ppm, followed by a reduction of $7.0$ ppm during the Bølling–Allerød (BA) period, and then an increase of $6.8$ ppm during the Younger Dryas (YD) period. However, the model underestimates the changes in atmospheric $pCO_2$ during these events compared to values derived from ice core data. Radiocarbon and stable isotope signatures ($\Delta^{14}C$ and $\delta^{13}C$) indicate that the model underestimates both the activated deep ocean ventilation and reduced efficiency of biological carbon export in the Southern Ocean, and the active ventilation in the North Pacific Intermediate Water during HS1. The relatively small changes in simulated atmospheric $pCO_2$ during HS1 might be attributable to these underestimations of ocean circulation variation. The changes in $\Delta^{14}C$ associated with strengthening and weakening of the AMOC during the BA and YD periods are generally consistent with values derived from sediment core records. However, although the data indicate continuous increase in $\delta^{13}C$ in the deep ocean throughout the YD period, the model shows the opposite trend. It suggests that the model either simulates excessive weakening of the AMOC during the YD period, or has limited representation of geochemical processes, including marine ecosystem response and terrestrial carbon storage. Decomposing the factors behind the changes in ocean $pCO_2$ reveals that variations in temperature and alkalinity have the greatest impact on change in atmospheric $pCO_2$. Compensation for the effects of temperature and alkalinity suggests that

the AMOC changes and the associated bipolar climate changes contribute to the decrease in atmospheric $p\text{CO}_2$ during the BA and increase in atmospheric $p\text{CO}_2$ during the YD period.

## 1   Introduction

Earth's climate has shifted from the colder conditions of the Last Glacial Maximum (LGM) to the warmer conditions of the Holocene. This climatic transition, known as the last deglaciation, occurred approximately 21 to 11 ka BP (thousand years before present). During this period, the atmospheric concentration of carbon dioxide ($p\text{CO}_2$) increased by almost 80 ppm (Barnola et al., 1987; Petit et al., 1999; Siegenthaler et al., 2005; Jouzel et al., 2007; Lüthi et al., 2008). The changes in the carbon cycle that affect the variation in atmospheric $p\text{CO}_2$ are closely related to the changes in climate observed during the last deglaciation.

In attempting to elucidate the mechanisms of climate change on the glacial–interglacial scale, previous modeling studies mainly focused on assessing the steady-state difference between the LGM and the preindustrial period. With the development of computational tools, transient climate modeling of the last glacial termination has recently been conducted, using temporal changes in insolation, greenhouse gas concentrations derived from ice core records, and meltwater fluxes from ice sheets (Lunt et al., 2006; Timm and Timmermann, 2007; Liu et al., 2009; He et al., 2013; Ganopolski and Roche, 2009; Menviel et al., 2011; Ivanovic et al., 2016; Obase and Abe-Ouchi, 2019; Obase et al., 2021; Kapsch et al., 2022; Bouttes et al., 2023). Transient climate modeling has distinct advantages because it avoids unrealistic equilibrium assumptions and it includes climate responses to internal variability or abrupt change. It also facilitates direct comparison between models and proxies, thereby allowing identification of temporal leads or lags in the process with respect to forcing. In these transient climate modeling studies, changes in atmospheric $p\text{CO}_2$, a greenhouse gas, are applied as external forcing. However, understanding the feedback between the climate and the carbon cycle is critical for understanding the long-term changes in climate dynamics. As fundamental research guided by this perspective, earlier modeling studies examined the temporal changes in the ocean carbon cycle during the last deglaciation or late Pleistocene, including glacial–interglacial cycles, using Earth System Models of Intermediate Complexity, e.g., CLIMBER-2 (Bouttes et al., 2012a; Brovkin et al., 2012; Mariotti et al., 2016; Ganopolski and Brovkin, 2017), Bern 3D (Tschumi et al., 2011; Menviel et al., 2012; Pöppelmeier et al., 2023) and LOVECLIM (Menviel et al., 2018; Stein et al., 2020).

Those earlier studies greatly advanced our understanding of changes in both the climate and the carbon cycle on scales of thousands to tens of thousands of years. However, the mechanisms behind the observed increase in atmospheric $p\text{CO}_2$ during the glacial termination are not fully understood. It has been suggested that release of carbon from the deep Southern Ocean to the atmosphere might have played a role in the rapid increase in atmospheric $p\text{CO}_2$ during the last deglaciation (Tschumi et al., 2011; Bouttes et al., 2012a; Mariotti et al., 2016; Menviel et al., 2018; Stein et al., 2020; Sigman et al., 2021; Gray et al., 2023; Sikes et al., 2023). The release of $\text{CO}_2$ can be triggered by disruption of the stratification in the Southern Ocean and changes

in the deep ocean circulation, which are affected by westerly winds in the Southern Hemisphere and brine rejection around Antarctica. In contrast, freshwater input from the Antarctic ice sheet increases stratification and does not lead to enhanced $CO_2$ outgassing. It has also been suggested that variation in North Pacific Intermediate Water (NPIW), associated with changes in the Atlantic Meridional Overturning Circulation (AMOC), might have contributed to the observed increase in atmospheric $pCO_2$ (Okazaki et al., 2010; Menviel et al., 2014).

To clarify the mechanisms behind long-term changes in ocean dynamics, the concentration or isotopic composition of elements in seawater can provide information on the processes responsible for their distribution. Thus, analysis of geochemical proxies in paleoclimatological archives can help oceanographers gain insight into past ocean variability and its underlying mechanisms.

Stable and radioactive carbon isotopes of dissolved inorganic carbon (DIC) in seawater are representative oceanic chemical tracers. In seawater, lighter carbon isotopes are preferentially taken up by phytoplankton during photosynthesis and they accumulate in the deep ocean as organic matter is degraded (O'Leary, 1981; Broecker and Maier-Reimer, 1992; Schmittner et al., 2013). Anomalies in the stable carbon isotope signature ($\delta^{13}C$) produced by the biological carbon pump spread globally in association with the deep ocean circulation. Therefore, the $\delta^{13}C$ value differs among water masses within the ocean interior. Radiocarbon ($^{14}C$) is introduced into seawater through gas exchange at the ocean surface, and it subsequently decreases in concentration through radioactive decay of $^{14}C$. Therefore, the radiocarbon isotope signature ($\Delta^{14}C$) serves as an indicator of the deep water flow rate (Stuiver et al., 1983).

Previous modeling studies examined the processes responsible for glacial–interglacial changes in the distribution of $\delta^{13}C$ and $\Delta^{14}C$ (Kurahashi-Nakamura et al., 2017; Menviel et al., 2017; Muglia et al., 2018; Wilmes et al., 2021; Kobayashi et al., 2021). It is generally assumed that deep water originating from the Southern Ocean with low $\delta^{13}C$ and $\Delta^{14}C$ expanded into the deep Atlantic Ocean during the LGM. It is proposed that the carbon isotope distribution during the LGM can be better explained by considering an effective biological pump associated with iron fertilization in the Southern Ocean and a shallower AMOC (Kobayashi et al., 2021).

Other proxies used to infer past water mass distribution and deep ocean circulation include the neodymium isotope ratio ($\varepsilon$Nd) and the protactinium–thorium ratio ($^{231}Pa/^{230}Th$). The $\varepsilon$Nd ratio can be used as a proxy for basin-scale water mass structure because the endmembers differ in each water mass source region (Lippold et al., 2012; Howe et al., 2016). The sedimentary $^{231}Pa/^{230}Th$ ratio, used as a proxy for change in flow rate, suggests that the strength of the AMOC might have changed markedly during the last deglaciation (McManus et al., 2004; Ng et al., 2018). Changes in the AMOC can influence the climatic state by altering not only the meridional interhemispheric heat transport, but also the ocean carbon cycle and associated changes in atmospheric $pCO_2$ (Schmittner and Galbraith, 2008; Menviel et al., 2008; Bouttes et al., 2012b). Nevertheless, there is ongoing debate regarding the magnitude and the direction of the change in atmospheric $pCO_2$ associated with AMOC variation (Gottschalk et al., 2019).

Several earlier modeling studies attempted to estimate AMOC variation by examining changes in the carbon isotope signature (Schmittner and Lund, 2015; Pöppelmeier et al., 2023). By estimating the AMOC that best fits the model and the data for $\delta^{13}C$ (Schmittner and Lund, 2015) or $\delta^{13}C$ and multiple chemical tracers in seawater (Pöppelmeier et al., 2023), those

studies improved our understanding of the relationship between changes in the AMOC and the ocean carbon cycle during the early deglaciation. In recent years, compilation of many sediment core records covering the last deglaciation has deepened our understanding of the spatiotemporal changes in carbon isotope ratios during this period (Zhao et al., 2018; Rafter et al., 2022; Muglia et al., 2023; Skinner et al., 2023). By comparing the compiled data with model output, it has been possible to gain valuable insights into deglacial changes in the ocean carbon cycle.

In this study, we conducted transient model experiments of the ocean carbon cycle and compared the model results with recently compiled sediment core records to investigate the mechanisms of deglacial changes in carbon isotope signatures. Additionally, we analyzed the drivers of the changes in atmospheric $pCO_2$ resulting from variations in the ocean carbon cycle. The objectives of this study were to clarify the reproducibility of deglacial carbon cycle changes in the model, understand the mechanisms underlying these changes, and identify the processes that are missing or underestimated in the model.

## 2 Methods

### 2.1 Model

In this study, numerical experiments on the ocean carbon cycle were performed using an offline ocean biogeochemical tracer model based on Parekh et al. (2005) within the framework of the CCSR Ocean Component Model version 4.0 (Hasumi, 2006). The model has an approximate horizontal resolution of 1 degree, and it includes 44 vertical layers with thicknesses ranging from 5 to 250 m.

The ocean biogeochemical cycle model is forced with monthly averaged output from a climate model simulation designed specifically for the last deglaciation and run with the MIROC 4m atmosphere–ocean general circulation model (AOGCM) (Obase and Abe-Ouchi, 2019; Obase et al., 2021). The boundary conditions include horizontal advection velocity, sea surface height, vertical diffusivity, temperature, salinity, shortwave radiation, wind speed above the sea surface, and sea ice concentration.

Prognostic variables for the ocean biogeochemical cycle model include phosphate, DIC, alkalinity, dissolved organic phosphate, dissolved oxygen, iron, silicate, and carbon isotopes of DIC ($^{13}$C and $^{14}$C). The availability of light, phosphate, and iron is used to determine the rate of phosphate uptake by phytoplankton. Notably, sedimentation processes on the seafloor are not considered, and all particles reaching the seafloor are assumed to dissolve in the deepest layer of the model.

### 2.2 Experimental design

To evaluate the transient response of the global ocean carbon cycle during the last deglaciation, we conducted offline ocean carbon cycle experiments forced with the outputs of the AOGCM MIROC 4m simulation, covering the period 21 to 11 ka BP. The MIROC 4m simulation focusing on the last deglaciation was performed according to the PMIP protocol (Ivanovic et al., 2016) with respect to the changes in orbital parameters and greenhouse gases during this period (Obase and Abe-Ouchi, 2019). However, the ice-sheets were fixed at their 21 ka BP state of the ICE-5G reconstruction. Freshwater inflow from the Northern

Hemisphere ice sheets deviates from the PMIP protocol after the latter half of Heinrich Stadial 1 (HS1). This approach seeks to align the simulated AMOC variations with those reconstructed from $^{231}\text{Pa}/^{230}\text{Th}$ sediment core records and the associated climate changes that occurred during the Bølling–Allerød (BA) and Younger Dryas (YD) periods.

### 2.2.1 Steady-state experiment on the ocean carbon cycle for the Last Glacial Maximum

The ocean biogeochemical cycle model was initialized through spin-up under the LGM ocean state (21 ka BP) calculated by the AOGCM. Dust deposition to the ocean surface was taken from a simulation conducted using the SPRINTARS aerosol transport–radiation model computed under LGM climatic conditions (Takemura, 2005). We assumed that iron deposition at the sea surface accounts for 3.5 wt% of the total dust deposition, with an assumed iron solubility of 1% (Parekh et al., 2005), which was derived from the ratio of wet and dry dust deposition and its solubility. However, it should be noted that some uncertainty is associated with these parameters. The initial distribution of ocean biogeochemical tracers was taken from the climatology of the World Ocean Atlas 2001 (Conkright et al., 2002; Locarnini et al., 2002) and the Global Ocean Data Analysis Project (Key et al., 2004). The initial iron concentration was set to a constant value of 0.6 nmol. The model is initialized with values of atmospheric $\delta^{13}\text{C}$ and $\Delta^{14}\text{C}$ of $-6.5‰$ and $0‰$ respectively.

Notably, the biogeochemical cycle simulation of the LGM ocean performed in this study did not include certain processes such as enhanced Southern Ocean stratification, iron fertilization from glaciogenic dust, and carbonate compensation, as discussed in Kobayashi et al. (2021). These three processes are found to contribute to the glacial reduction in atmospheric $p\text{CO}_2$, with values of 294.7 ppm for the preindustrial run (PI_sed in Kobayashi et al. (2021)) and 217.4 ppm for the LGM run (LGM_all in Kobayashi et al. (2021)). Therefore, the calculated atmospheric $p\text{CO}_2$ during the LGM is expected to be higher than the atmospheric $p\text{CO}_2$ reported in Kobayashi et al. (2021). Further experiments with the ocean general circulation model are needed to obtain a physical ocean field that accounts for the enhanced stratification of the Southern Ocean during glacial periods. Therefore, it was not possible to include this process in the model settings adopted for this study.

### 2.2.2 Transient experiment on the ocean carbon cycle during the last deglaciation

A transient experiment was conducted to investigate the ocean carbon cycle during the last deglaciation, starting from the initial state of the LGM (21 ka BP). Figure 1 illustrates the temporal variations of temperature and AMOC of the AOGCM output imposed as forcing of the ocean biogeochemical cycle model. During the transient experiment, dust deposition was periodically adjusted every 100 years based on the scaling between the LGM and the Holocene, using the reconstructed dust deposition from the Dome Fuji ice core (Dome Fuji Ice Core Project Members, 2017). Notably, the transient experiment did not account for temporal variations in ocean volume caused by ice sheet changes and associated changes in mean ocean concentrations of biogeochemical tracers (i.e., nutrients, alkalinity, and DIC) (Lhardy et al., 2021). This simplification must be revisited in future studies.

Upon completion of the LGM ocean spin-up, we established a restoring term to counteract drifts in $\delta^{13}\text{C}$ and $\Delta^{14}\text{C}$ caused by gas exchange between the atmosphere and the ocean. This restoring term includes the exchange of carbon isotopes between

the atmosphere and the land, together with the production of $^{14}$C in the atmosphere. This restoring term was assumed constant throughout the deglaciation experiment.

## 3   Results

### 3.1   Ocean carbon cycle state during the LGM

Figure 2 shows the calculated variations in $\Delta^{14}$C–$CO_2$, $\delta^{13}$C–$CO_2$, and $pCO_2$ in the atmosphere (solid lines), together with their estimated values (dashed lines). Additionally, it illustrates the variations in the average of $\Delta\Delta^{14}$C (i.e., the difference in $\Delta^{14}$C between the ocean and the atmosphere), $\delta^{13}$C, and DIC in the middle (500–2000 m) and deep ($> 2000$ m) layers of the Atlantic, Southern ($< 40°$S), and Pacific oceans.

Analysis of the modeled differences between the LGM and the Holocene indicates that the AMOC is weaker (9.0 Sv) during
the LGM (21 ka BP) than during the Holocene (11 ka BP), i.e., 17.4 Sv (Figs. 1a and S1). The basin-wide distributions of $\Delta\Delta^{14}$C and $\delta^{13}$C in the Atlantic and Pacific oceans during specific periods are presented in Figs. 3 and 4, respectively. The model results demonstrate that $\Delta\Delta^{14}$C, an indicator of ocean ventilation, is lower in the deep Atlantic at 21 ka BP than at 11 ka BP (Figs. 2c, 3a, and 3f), which is in qualitative agreement with reconstructions from sediment core records (Rafter et al., 2022). Notably, however, the simulated $\Delta\Delta^{14}$C values are less negative than the reconstruction $\Delta\Delta^{14}$C values at 21 ka BP in
the Atlantic below 3000 m and in the Pacific below 2000 m (Fig. 3a).

Regarding $\delta^{13}$C, the model results show a stronger vertical gradient between the surface and the deep ocean in the Atlantic (Fig. 2h), corresponding to a shallower and weaker AMOC at 21 ka BP compared to that at 11 ka BP. This qualitative difference is consistent with the reconstructed $\delta^{13}$C. However, similar to the finding for $\Delta\Delta^{14}$C, our model experiment underestimates the reconstruction of $\delta^{13}$C in the deep ocean, particularly in the Southern Ocean (Fig. 2i).

Our previous study, which involved numerical experiments under the climatic conditions of the LGM, and accounted for the enhanced stratification in the Southern Ocean and iron fertilization from glaciogenic dust, showed improved quantitative agreement between the model results and the sediment core data for dissolved oxygen, $\delta^{13}$C, and $\Delta^{14}$C (Kobayashi et al., 2021). The triangles in Fig. 2 illustrate the changes in the carbon isotope signatures reported in that study. Comparison of the results from the two studies highlights the advances made by Kobayashi et al. (2021) in capturing the dynamics of the South-
ern Ocean, suggesting that incorporation of the processes considered in their research could improve model–data agreement. However, their LGM simulation also slightly overestimated the changes in the glacial Pacific, and these discrepancies highlight the difficulty of achieving consistent scenarios that account for all changes in the global ocean within a model.

Atmospheric $pCO_2$ was predicted by running the ocean carbon cycle model. Its value is 278.1 ppm at 21 ka BP and 306.9 ppm at 11 ka BP, i.e., a difference of 28.8 ppm that is relatively small compared to the difference of approximately
80 ppm reconstructed from ice cores (approximately 188–267 ppm in the EPICA Dome C record (Bereiter et al., 2015) and approximately 193–273 ppm in the WAIS Divide record (Bauska et al., 2021)). This discrepancy could be attributed to several factors, including the relatively small differences in SST observed between the two intervals (Fig. 1b). Using proxy and data assimilation, the global mean SST difference between the LGM and the Holocene has been reported to be 1.7–3.6 °C

(MARGO Project Members, 2009; Tierney et al., 2020; Paul et al., 2021; Annan et al., 2022), whereas the SST difference in our experiment is only 1.6 °C. A small difference in SST leads to a small difference in $CO_2$ solubility between the two periods, which causes underestimation of the magnitude of atmospheric $pCO_2$. Furthermore, it is important to note that this study did not consider specific processes that might have contributed to the reduction in atmospheric $pCO_2$ during the LGM, such as enhanced salinity stratification, iron fertilization from glaciogenic dust, and carbonate compensation, as discussed in Kobayashi et al. (2021).

The differences in the steady-state ocean carbon cycles at 21 and 11 ka BP highlight the difficulty in accurately reproducing the actual changes in atmospheric $pCO_2$ in the transient experiment connecting these periods. Therefore, our analysis focuses on investigation of the impacts of climate change, particularly the notable variations in the AMOC, on the ocean carbon cycle, and reveals the successes and deficiencies of the model through model–data comparison of carbon isotope signatures.

## 3.2 Carbon isotope changes during the last deglaciation

### 3.2.1 Radiocarbon isotopes

Here, we present the calculated transient changes in the ocean carbon cycle during the last deglaciation. During the deglaciation, atmospheric $\Delta^{14}C$ decreases from approximately $500‰$ to $0‰$ (Reimer et al. (2020); Fig. 2a). Seawater $\Delta\Delta^{14}C$ generally increases from the relatively low LGM values, but decreases during HS1 and the YD period (Fig. 2b–e). Generally, the state of the AMOC strongly influences the distribution of $\Delta\Delta^{14}C$. Both the model and the sediment core records show increase in $\Delta\Delta^{14}C$ in the deep ocean during periods when the AMOC is relatively strong, e.g., the BA period and the Holocene (Fig. 3d and 3f), and present decrease in $\Delta\Delta^{14}C$ during periods when the AMOC is relatively weak, e.g., HS1 and the YD period (Fig. 3b and 3c). The calculated changes in $\Delta\Delta^{14}C$ are consistent with the pattern observed in the sediment core record during a period characterized by rapid change in the AMOC after the BA transition (14.7 ka BP).

In the Pacific, when the AMOC is strong (i.e., at 13 and 11 ka BP), the $\Delta\Delta^{14}C$ in the South Pacific is relatively high, with elevated values from the surface to the deep Southern Ocean along the path of the AABW. The sediment core records compiled in Rafter et al. (2022) suggest elevated $\Delta\Delta^{14}C$ values in the North Pacific intermediate layers during HS1, possibly indicating the influence of the NPIW intrusion. However, the model results do not provide clear indication of the intrusion of young water masses (Figs. 3 and S4).

The compiled sediment core records globally show substantial increase in $\Delta\Delta^{14}C$ during HS1. In contrast, the model experiment does not show such pronounced change (Fig. 2b–e). This discrepancy can be attributed to two main factors: failure of the model experiment to simulate activation of ocean ventilation during HS1, and the greater magnitude of the initial $\Delta\Delta^{14}C$ values at 21 ka BP relative to the reconstructed values. This is supported by the insufficient carbon sequestration in the ocean calculated during the LGM (Fig. 2k). In other words, considering the glacial–interglacial redistribution of carbon in the atmosphere–ocean system, the relative abundance of $^{14}C$ to $^{12}C$ in the atmosphere is higher during the ice age, as manifested in the atmospheric $\Delta^{14}C$–$CO_2$ (Fig. 2a); however, the variation in $^{12}C$ is not well reproduced in the model experiment (Fig. 2k).

Regarding the latter point of insufficient carbon sequestration during the LGM, the triangles shown in Fig. 2 represent the results of the best LGM simulation (LGM_all) conducted by Kobayashi et al. (2021). That simulation incorporated enhanced salinity stratification and sedimentation processes, which further contribute to accurate reproduction of low $\Delta\Delta^{14}$C of deep water during the LGM. As shown in Fig. 2b–e, there is substantial discrepancy between the model and the reconstruction, particularly in relation to the Southern Ocean during the period of early deglaciation. Incorporation of change in vertical mixing resulting from variation in ocean stratification could potentially improve the simulation of $\Delta^{14}$C in the deglaciation.

### 3.2.2 Stable carbon isotopes

Next, we focus on the changes in $\delta^{13}$C. For seawater $\delta^{13}$C, the overall trends of change in $\delta^{13}$C and $\Delta\Delta^{14}$C are similar and correspond to phases of climatic change; however, $\delta^{13}$C is less sensitive than $\Delta\Delta^{14}$C to climate change (Fig. 2). This differential response might be related to biological fractionation of carbon isotopes. A more active AMOC leads to increased biological activity that reduces $\delta^{13}$C in deeper layers, especially in the North Atlantic. This counteracts the influence of lighter carbon transported to the surface by the active AMOC.

During HS1, $\delta^{13}$C decreases gradually in the upper 3000 m of the North Atlantic (Figs. 2h, 4, and S3). The reduction in $\delta^{13}$C can be attributed to several factors (Gu et al., 2021) that include increased contribution from southern-sourced deep water with low $\delta^{13}$C endmembers, accumulation of remineralized carbon with low $\delta^{13}$C attributable to a weakened AMOC and reduced ventilation, and potential increase in the $\delta^{13}$C endmember of North Atlantic Deep Water (NADW). The results of this study show no clear change in the NADW endmembers of $\delta^{13}$C (Fig. S5). Therefore, the change in $\delta^{13}$C is attributed to weakened ventilation in the North Atlantic and to expansion of southern-sourced deep water. However, the observed $\delta^{13}$C change is relatively small compared to that derived from sediment core data because the AMOC change is less pronounced than that expected from the $^{231}$Pa/$^{230}$Th reconstruction (McManus et al., 2004; Ng et al., 2018).

The deep Southern Ocean has its lowest $\delta^{13}$C during the LGM, although the value gradually increases during HS1. However, these observed changes are not reproduced in the model. According to Kobayashi et al. (2021), the low $\delta^{13}$C during the LGM is related to enhanced Southern Ocean stratification and iron fertilization from glaciogenic dust. These processes, which are not considered in this study, contribute to the differences between the model and the observed data.

During the BA period (Fig. 4d) and the Holocene (Fig. 4f), the intensified and deepened AMOC contributes to high $\delta^{13}$C values originating from the North Atlantic penetrating to depths below 2000 m. Basin-averaged $\delta^{13}$C reconstructions also show this increase in $\delta^{13}$C, especially in the Atlantic (Fig. 2g–j). However, in contrast to the calculated change, the sediment core records do not show further reduction in $\delta^{13}$C in the deep ocean during the YD period.

Changes in $\delta^{13}$C in the deep ocean lead to changes in atmospheric $\delta^{13}$C–CO$_2$. There is a sharp drop in atmospheric $\delta^{13}$C–CO$_2$ during HS1, followed by slight rise during the BA period and then further decline during the YD period (Schmitt et al. (2012); Fig. 2f). However, the model-calculated changes in the ocean carbon cycle do not reproduce this trend in atmospheric $\delta^{13}$C–CO$_2$. The reconstructed $\delta^{13}$C–CO$_2$ increases during the BA period, decreases during the YD period, and then increases again, whereas the model-calculated trend is the opposite. The discrepancy might involve the contribution from changes in vegetation, which is a topic discussed in Section 4.3.

Isotope fractionation through temperature-dependent gas exchange and phytoplankton preference for uptake of lighter carbon also play important roles in $\delta^{13}$C variation. Figure S6 shows the calculated changes in organic carbon export from that in 21 ka BP with qualitative changes in biological flux reconstructed from proxies, specifically opal flux and alkenone flux, in sediment core records (Chase et al., 2003; Anderson et al., 2009; Bolton et al., 2011; Kohfeld and Chase, 2011; Martínez-García et al., 2014; Maier et al., 2015; Studer et al., 2015; Thiagarajan and McManus, 2019; Ai et al., 2020; Weber, 2021; Li et al., 2022). During HS1, both the model and the proxies show increased biological carbon transport in the polar region of the Southern Hemisphere (Fig. S6a–d). In the polar regions, sea ice is reduced owing to warming, which could result in less light limitation and allow increased biological productivity. However, biological carbon transport is reduced in subpolar regions and in the South Pacific gyres. These changes in southern regions can be attributed to reduced nutrient supply resulting from weakening of the AMOC. Another important factor is the increase in iron limitation associated with the reduced supply of dust-derived iron that affects biological production. During the BA warm period, the enhanced AMOC enhances nutrient transport from the deep to the surface ocean, resulting in increased biological transport in the North Atlantic (Fig. S6e and 6f). These changes in the vertical nutrient transport then propagate to the North Pacific. From the BA period to the YD period, there is increase in biological export in the Southern Ocean that might be attributable to reduction in sea ice resulting from warming in the Southern Hemisphere. The factors that alter biological production in the model are understood but need to be constrained using additional proxy data with high temporal resolution that can capture millennial-scale variations.

### 3.3 Deglacial changes in atmospheric $p$CO$_2$ caused by changes in the ocean carbon cycle

Figure 2k shows the calculated changes in atmospheric $p$CO$_2$ driven by variations in the climate and the carbon cycle during the last deglaciation. To investigate the factors driving the change in atmospheric $p$CO$_2$, we decomposed the factors relevant to the partial pressure of CO$_2$ at the sea surface ($p$CO$_2{}^{os}$). This parameter controls atmospheric $p$CO$_2$ through gas exchange between the atmosphere and the ocean. Oceanic $p$CO$_2$ is affected by temperature, salinity, DIC, and alkalinity, and the influence of those factors on $p$CO$_2{}^{os}$ can be represented as follows:

$$p\text{CO}_2{}^{os} = f(\text{sDIC}, \text{sALK}, \text{SST}, \text{SSS}) \tag{1}$$

where sDIC is sea surface DIC, sALK is sea surface alkalinity, SST is sea surface temperature, and SSS is sea surface salinity. The function $f$ is determined based on the inorganic chemistry of the carbonate system (Millero, 1995). We can assess the contribution of each variable to the changes in $p$CO$_2{}^{os}$ by examining the change in each variable from its original value.

### 3.3.1 Heinrich Stadial 1

During HS1, the calculated atmospheric $p$CO$_2$ rises slightly until approximately 17 ka BP, and then it rises sharply to approximately 15 ka BP. From 18 to 15 ka BP, atmospheric $p$CO$_2$ increases by $10.2$ ppm (Fig. 5a), whereas the WAIS Divide ice core record shows increase of $41.4$ ppm during the same period (Bauska et al., 2021). The model accounts for approximately one quarter of the reconstructed changes in atmospheric $p$CO$_2$. Decomposition analysis of $p$CO$_2{}^{os}$ reveals that most of the variation in $p$CO$_2{}^{os}$ is driven by change in SST (Fig. 5a and 5b). The changes in $p$CO$_2{}^{os}$ ($\Delta p$CO$_2{}^{os}$) attributable solely to

variations in temperature and salinity ($\Delta p CO_2^{os}$(TS)) and in DIC and alkalinity ($\Delta p CO_2^{os}$(CA)) are shown in Fig. 6a and 6b, respectively. It is evident that $\Delta p CO_2^{os}$(TS) shows a predominantly positive contribution globally, reflecting the pattern of SST increase (Fig. 6d), because increasing SST reduces $CO_2$ solubility. In other words, the main contributor to the increase in 290 $p CO_2^{os}$ during HS1 is warming, especially in the subantarctic region.

### 3.3.2 Bølling-Allerød period

At the onset of the BA transition near 14.7 ka BP, atmospheric $p CO_2$ begins to decrease, and this reduction continues until 12.8 ka BP (Fig. 2k). From 15 to 13 ka BP, atmospheric $p CO_2$ decreases by 7.0 ppm (Fig. 5c). During the BA period, the contributions of thermal changes and biogeochemical changes act in opposition to the change in atmospheric $p CO_2$ (Fig. 295 5c and 5d). At the onset of the BA period, as the AMOC strengthens (Fig. 1a), SST and SSS both increase in the Northern Hemisphere and decrease in the Southern Hemisphere (Fig. 7d). The net contribution of $p CO_2^{os}$(TS) attributable to changes in $CO_2$ solubility is positive (Fig. 5c and 5d). However, the enhanced AMOC facilitates the transport of nutrients, carbon, and alkalinity from the deep to the surface ocean, especially in the North Atlantic (Fig. 7e and 7f). The increase in sDIC leads to increase in $p CO_2^{os}$, while the increase in sALK leads to reduction in $p CO_2^{os}$. These opposing effects partially offset 300 each other, resulting in net reduction in $p CO_2^{os}$ (Figs. 5c, 5d, and 7c). During the AMOC overshoot at the BA transition and the subsequent stabilized phase, increased biological production in most of the global ocean contributes to millennial-scale decrease in sDIC, resulting in reduction in $p CO_2^{os}$ (Fig. 5c and 5d). In summary, following the recovery of the AMOC, the opposing contributions of $\Delta p CO_2^{os}$(TS) and $\Delta p CO_2^{os}$(CA) to $\Delta p CO_2^{os}$ over time control the temporal changes in $p CO_2^{os}$ and subsequent reduction in atmospheric $p CO_2$.

### 305 3.3.3 Younger Dryas period

Atmospheric $p CO_2$ rises again at the onset of the YD period (12.8 ka BP), coinciding with the collapse of the AMOC into a weak state (Fig. 1a). From 13 to 12 ka BP, atmospheric $p CO_2$ increases by 6.8 ppm (Fig. 5e). Decomposition analysis of $p CO_2^{os}$ reveals that the influences of $\Delta p CO_2^{os}$(TS) and $\Delta p CO_2^{os}$(CA) on the overall $\Delta p CO_2^{os}$ are in opposition, and this offset is also observed during the BA period, but in the opposite sense. The contribution of $\Delta p CO_2^{os}$(TS) increases over 310 time, leading to increase in $p CO_2^{os}$ (Fig. 5e and 5f). The contribution of $\Delta p CO_2^{os}$(CA) is small. As the AMOC weakens, a decrease in sALK contributes to an increase in $p CO_2^{os}$, while a decrease in sDIC contributes to a decrease in $p CO_2^{os}$ (Figs. 5e, 5f, 8c, 8e, and 8f). However, the net effect of changes in DIC and alkalinity on $p CO_2^{os}$ is minimal, resulting in only a slight decrease in $\Delta p CO_2^{os}$(CA) during the YD period. From these opposing effects, the overall changes in $\Delta p CO_2^{os}$(TS) and $\Delta p CO_2^{os}$(CA) indicate an increase in $\Delta p CO_2^{os}$ during the YD period.

### 315 4 Discussion

The objective of this study was to investigate the transient response of the ocean carbon cycle during the last deglaciation. By comparing the calculated carbon isotope signatures of $\delta^{13}C$ and $\Delta^{14}C$ with those derived from sediment core records, we can

assess the impacts of changes in climate and the AMOC on those signatures. This comparison can also present information to help identify potential biases or missing processes within the model. In addition to changes in atmospheric $p\mathrm{CO}_2$, investigating the mechanisms behind changes in carbon isotopes contributes to a more comprehensive understanding of the temporal changes in the global carbon cycle.

### 4.1 Response of carbon isotope signatures to drastic changes in the deep ocean circulation

Comparison of $\Delta\Delta^{14}\mathrm{C}$ variations between models and data enables assessment of the accuracy of calculated ocean circulation changes. The reconstructed $\Delta\Delta^{14}\mathrm{C}$ in the deep ocean rises notably during the latter half of HS1 (Rafter et al., 2022), in contrast to the less pronounced shift seen in the model experiment (Fig. 2b–e). Two primary factors contribute to this discrepancy. First, the model underestimates the increase in deep ocean ventilation during the latter half of HS1 period, which is crucial for determining the trend in $\Delta^{14}\mathrm{C}$ changes in the deep ocean. Second, the model calculates higher $\Delta\Delta^{14}\mathrm{C}$ values compared to the reconstructed lower values of $\Delta\Delta^{14}\mathrm{C}$ in the deep ocean during the LGM. These aspects indicate a potential oversight in the model's representation of ocean dynamics, affecting both the simulation of ventilation changes during the latter half of HS1 and $\Delta\Delta^{14}\mathrm{C}$ values during the LGM. The difficulty in reproducing changes in $\Delta\Delta^{14}\mathrm{C}$ might also be related to underestimation of variations in atmospheric $p\mathrm{CO}_2$ during this period (Fig. 2k). Processes that might contribute to this problem are discussed in more detail in Section 4.2.

After the BA transition, the calculated variations in $\Delta\Delta^{14}\mathrm{C}$ in synchrony with the significant changes in the AMOC are generally consistent with those observed in the reconstruction (Rafter et al., 2022). Corresponding to the AMOC change, $\Delta\Delta^{14}\mathrm{C}$ increases in the deep ocean from the Atlantic to the Pacific Ocean (Fig. 2c–e). Subsequently, $\Delta\Delta^{14}\mathrm{C}$ decreases from the Atlantic to the Southern Ocean during the YD period in response to weakening of the AMOC (Fig. 2c–e). This change is consistent with the reconstruction (Rafter et al., 2022), but there is an overestimation of the quantitative changes in the deep Atlantic Ocean, as illustrated in Fig. 3e. This overestimation may be related to the challenges in accurately reproducing the deep ocean circulation fields, which is a topic that is discussed further below.

Assessment of the AMOC changes during the last deglaciation by Pöppelmeier et al. (2023) involved conducting transient model simulations using the Bern3D model. They performed multiple model–data comparisons including carbon isotope ratios, $\varepsilon\mathrm{Nd}$, and $^{231}\mathrm{Pa}/^{230}\mathrm{Th}$. Their research results suggest gradual weakening of the AMOC during HS1, recovery at the BA transition, and subsequent weakening during the YD period, albeit without a complete collapse. The proposed pattern of AMOC change is qualitatively consistent with the ocean modeling of Obase and Abe-Ouchi (2019) (Fig. 1a). The Bern3D study indicates that the proportion of deep water originating from the North Atlantic during the YD period is little different to that during the BA because of the short duration of the YD period. Conversely, this study shows drastic changes in the distribution of $\Delta^{14}\mathrm{C}$ and $\delta^{13}\mathrm{C}$ at the basin scale in response to AMOC variations over a period of approximately $1000\,\mathrm{years}$ (Figs. 3e and 4e). This extended period of the AMOC stagnation in our model helps to explain the observed discrepancies between the model and the reconstructed data (Rafter et al., 2022; Muglia et al., 2023) in the deep Atlantic during the YD period.

The carbon isotope ratios calculated by our model suggest that the AMOC during the YD period might be represented as excessively weak or that the duration of the weak AMOC state may be overly extended. However, it is important to acknowledge the inherent differences among models in representing broader deep ocean circulation patterns, including the AABW and Pacific meridional overturning. These differences result in distinct chemical tracer distributions at the basin scale, underscoring the challenge of conclusively explaining past AMOC variations through a single model or study. Given the systematic biases present in physical and biogeochemical processes within models, a comparative analysis across multiple models is essential for exploring past AMOC variations.

Further information can be obtained by comparing the results of model–data comparisons for $\Delta\Delta^{14}C$ and $\delta^{13}C$. For $\delta^{13}C$, both the model and the data show similar trends, depicting increase in deep water $\delta^{13}C$ during the BA period as in $\Delta\Delta^{14}C$. However, there is a discrepancy during the subsequent YD period. The model indicates reduction in deep water $\delta^{13}C$ during the YD period, whereas this feature is absent in the reconstruction of Muglia et al. (2023) (Fig. 2g–j). There are several possible factors that could potentially influence this discrepancy. For example, discrepancies in the directions and magnitudes of changes observed across different sediment cores could reflect inherent variability in environmental signals. Additionally, potential dating inaccuracies within individual sediment core data could result from smoothing effects such as bioturbation and coring artifacts. These complexities in interpreting the sediment core record stems from both natural variability and methodological challenges, highlighting the need for caution when comparing model simulations with sediment core data. Another important factor is the weakening of the simulated AMOC during the YD period. Comparison of the model and sediment data for $\Delta\Delta^{14}C$ and $\delta^{13}C$ suggests that the weakening of the AMOC during the YD period might be overly pronounced in the model (Figs. 3e and 4e). Additionally, the calculated increase in export of biogenic organic matter in the Southern Ocean during the YD period compared to that in the BA period (Fig. S6f and S6g) contributes to the decrease in $\delta^{13}C$ in the deep ocean. This emphasizes the importance of accurately simulating nutrient and iron cycles, especially in iron-limited regions affected by changes in dust-derived iron supply. As the Southern Hemisphere warms and becomes more humid, the supply of iron from dust might decrease (Martin, 1990; Martínez-García et al., 2014). Reproducing changes in $\delta^{13}C$ is challenging owing to the intricate interconnections between ocean circulation, biological processes, and atmosphere–ocean gas exchange. Understanding the discrepancies between the model and the data in terms of the $\delta^{13}C$ changes will require future sensitivity experiments to clarify their respective contributions and to provide deeper understanding of these factors.

### 4.2 Insights from carbon isotope ratios: oceanic $CO_2$ release during the deglaciation

Ocean modeling with freshwater forcing experiments has provided insights into the link between the shutdown and resumption of the AMOC and the changes in atmospheric $pCO_2$. Schmittner and Galbraith (2008) demonstrated that cessation of the AMOC causes increase in atmospheric $pCO_2$ owing to several factors. First, the efficiency of the biological carbon pump in the North Atlantic is relatively high compared to that in the Southern Ocean. Therefore, reduction in the NADW inflow leads to decrease in biological carbon sequestration in the deep ocean. Second, weakening of Southern Ocean stratification associated with shutdown of the AMOC increases the outgassing of $CO_2$ from the ocean to the atmosphere.

The results of this study confirm the gradual increase in atmospheric $pCO_2$ during HS1 and the YD period in parallel with the weakened state of the AMOC; however, such an increase is not directly related to reduction in the regenerated nutrient inventory, as suggested by Schmittner and Galbraith (2008). The reason for this difference is that the contribution of the changes in temperature and alkalinity to the change in $pCO_2^{os}$ during the YD period is greater than the contribution of the change in DIC in this study. When the AMOC changes, the non-thermal effects on changes in $pCO_2^{os}$ mainly depend on the magnitude of the relative contributions of DIC and alkalinity to $pCO_2^{os}$, based on the vertical gradient of DIC and alkalinity between the surface and the deeper ocean (Figs. S7 and S8).

Although our results show the impact of drastic changes in the AMOC on atmospheric $pCO_2$ during the last deglaciation, the model does not fully explain the variations in atmospheric $pCO_2$ during the early deglaciation. Ice core records indicate a rise of approximately 40 ppm in atmospheric $pCO_2$ accompanied by reduction in atmospheric $\delta^{13}$C–$CO_2$ and $\Delta^{14}$C–$CO_2$ during HS1. However, the calculated variations are insufficient in terms of their amplitude (Fig. 2a, 2f, and 2k). A key contributor to this discrepancy is the limited extent of the change in ventilation, evident from the $\Delta\Delta^{14}$C changes in the ocean (Fig. 2b–e). However, the current model has difficulty fully reproducing these substantial changes in ventilation.

In addition to the changes in ventilation, biological processes are important in explaining the deglacial carbon cycle changes. Kobayashi et al. (2021) indicated that iron fertilization from glaciogenic dust increases biological production in the subantarctic region, thereby contributing to the reproduction of low $\delta^{13}$C in the deep Southern Ocean during the LGM (triangles in Fig. 2i). However, their study did not reproduce the profoundly old deep water in the deep glacial Southern Ocean, as suggested by radiocarbon data (triangles in Fig. 2d), but it did reproduce the low $\delta^{13}$C values (triangles in Fig. 2i). The change in glaciogenic dust deposition was not considered in this study, and therefore the model might underestimate the glacial–interglacial variation in biological production in the subantarctic region, potentially contributing to the underestimation of the changes in atmospheric and deep-sea $\delta^{13}$C during HS1.

Moreover, while many studies focused primarily on environmental changes in the Atlantic, the contributions from other ocean basins are also important. For example, sediment core records from the North Pacific indicate increase in $\Delta\Delta^{14}$C at depth near $1000\ \mathrm{m}$ during HS1 (Okazaki et al., 2010; Rae et al., 2014; Rafter et al., 2022). Analysis of radiocarbon and boron isotopes in sediment cores by Rae et al. (2014) revealed that the extent of the NPIW expanded during HS1. These ventilation changes have potential to contribute to the rise in atmospheric $pCO_2$. Chikamoto et al. (2012) compared two coupled climate models, i.e., MIROC version 3.1 and LOVECLIM, and showed consistent results for activation of ventilation of NPIW triggered by freshwater inflow into the North Atlantic. The processes of activation of ocean ventilation and subsequent degassing of $CO_2$ in the North Pacific during the early deglaciation could contribute to the currently unexplained increase in atmospheric $pCO_2$.

The high-resolution ice core data obtained from the WAIS Divide record provides valuable insights into the time scale of the changes in the carbon cycle during the last deglaciation. It is suggested that there are two modes of change in relation to atmospheric $pCO_2$: slow increase on the millennial scale and rapid increase on the centennial scale (Marcott et al., 2014). The rapid increase in atmospheric $pCO_2$ of 10–15 ppm at the end of HS1 (14.8 ka BP) and at the end of the YD period (11.7 ka BP) over a short period of 100–200 years is synchronized with the resumption of the AMOC (McManus et al., 2004; Ng et al., 2018). However, this study did not reproduce such abrupt changes in atmospheric $pCO_2$. Several factors might contribute to

this discrepancy, including insufficient temperature rise in the Southern Hemisphere associated with change in the AMOC, inadequate representation of the vertical concentration gradients of DIC and alkalinity, limitations in capturing atmospheric and oceanic dynamics in the general circulation model, and the influence of small-scale phenomena. Previous modeling studies have suggested that deeper convection in the Southern Ocean and strengthening of westerly winds in the Southern Hemisphere could contribute to the abrupt jump in atmospheric $p\mathrm{CO}_2$ during the middle of HS1 (16.3 ka BP) by transporting sequestered carbon from the deep Southern Ocean to the surface (Menviel et al., 2018). These processes are related to the challenges in reproducing carbon isotope ratios described above; therefore, this discussion points to the necessity of improving our AOGCM and of refining its experimental setup in future studies, as previewed in Section 4.3.

## 4.3 Improvement of the model and the experimental design: future implications

Model–data comparisons of carbon isotope signatures underscore the importance of refining our climate models to more accurately represent the complex interactions that govern changes in the carbon cycle. Future improvements to the AOGCM used in this study might address the following considerations.

Currently, the AOGCM does not account for temporal changes in ice sheets and it underestimates the changes in Southern Ocean SST (Obase and Abe-Ouchi, 2019). Moreover, there is some uncertainty regarding the volume of meltwater flow across the North Atlantic during HS1 (Ivanovic et al., 2018; Snoll et al., 2023) and the BA period (Kapsch et al., 2022; Bouttes et al., 2023). It suggests that the problem is integral to the AOGCM because the AMOC response is not consistently and realistically observed, even when realistic freshwater variations are applied. Furthermore, as identified by Obase et al. (2023), the magnitude of ocean warming during the last deglaciation varies depending on the response characteristics of each model, resulting in a range across multiple models. Sherriff-Tadano et al. (2023) demonstrated that alteration of parameters associated with cloud thermodynamic phase fractions in a climate model reduces the warming bias of SST in the modern Southern Ocean. Using a model with a reduced Southern Ocean warming bias, we expect to obtain different responses in Southern Ocean SST and ocean circulation during the glacial period, and in their changes during the deglaciation, compared to those derived in this study. Some studies proposed potential alterations in the westerly winds over the Southern Ocean throughout the last deglaciation (Gray et al., 2023), but there is a substantial degree of uncertainty concerning the anticipated changes in the atmospheric dynamics. Those uncertainties could have substantial impact on the results of ocean biogeochemical cycle modeling and, consequently, on atmospheric $p\mathrm{CO}_2$. Efforts to reduce bias and to facilitate comprehensive discussion regarding climate model consistency are critical to advancing future climate–carbon cycle modeling. These endeavors are essential to refine our understanding of the complex interactions between climate and the carbon cycle.

In addition to those factors mentioned above, there are several other factors that could contribute to improving the simulation of the ocean carbon cycle during the last deglaciation. One important consideration is the inclusion of critical processes for lowering atmospheric $p\mathrm{CO}_2$ during the LGM, which are identified in Kobayashi et al. (2021). Those processes include enhanced stratification of the Southern Ocean, iron fertilization from glaciogenic dust, and carbonate compensation. Understanding the changes in those processes during the deglaciation is critical, and their proper incorporation into future modeling efforts might lead to both improved simulations and better understanding of the dynamics during this period. Studies have also reported that

inclusion of the parameterization of vertical mixing, which depends on tidal mixing energy and stratification, could help better reproduce the deep ocean circulation in the Pacific (Oka and Niwa, 2013; Kawasaki et al., 2022). The introduction of tidal

mixing parameterization has also proven effective in reproducing $\delta^{13}$C and $\Delta^{14}$C in the glacial ocean (Wilmes et al., 2021). Incorporating the insights from these model developments has the potential to lead to more realistic representation of carbon cycle variations during the glacial period and subsequent deglaciation. Moreover, glacial–interglacial changes in ocean volume due to ice sheet changes also have an impact on the carbon cycle. A recent study analyzing PMIP model outputs highlighted the importance of accurate representation of ocean volume changes and their associated effects on alkalinity adjustments (Lhardy

et al., 2021). For more accurate simulations, it is critical to perform numerical integration that accounts for temporal changes in ocean volume during the deglaciation.

It is also worth noting that carbon exchange between the atmosphere and the ocean is not the sole driver of deglacial variation in atmospheric $p$CO$_2$. Changes in terrestrial and soil carbon storage also play important roles in modulating atmospheric $\delta^{13}$C–CO$_2$ and $p$CO$_2$ during the last deglaciation (Schmitt et al., 2012; Jeltsch-Thömmes et al., 2019; Bauska et al., 2021).

Comparison of $\Delta^{14}$C–CO$_2$ and $\delta^{13}$C–CO$_2$ provides a consistent explanation if carbon uptake by vegetation expands during the BA period and declines during the YD period, as suggested by Schmitt et al. (2012). Vegetation growth, prompted by CO$_2$ fertilization, can act as a carbon sink, offsetting the increase in atmospheric $p$CO$_2$ (Bouttes et al., 2012a; Menviel et al., 2012). In this study, we applied the same sea surface restoring terms for the carbon isotopes used during the initial spin-up of the LGM throughout the deglaciation experiment. In other words, this approach did not consider changes in vegetation or changes in $^{14}$C

production in the atmosphere. Future studies using Earth system models that include both terrestrial and oceanic carbon cycle processes would enhance our comprehensive understanding of glacial changes in carbon cycles. A study of the carbon cycle associated with the glacial Dansgaard–Oeschger events, conducted using an earth system model, revealed that the changes in terrestrial carbon storage at this time scale are as important as those in the oceans (Jochum et al., 2022). Moreover, in previous studies using Earth System Models of Intermediate Complexity, temporal variations in atmospheric $p$CO$_2$ are primarily used

to calculate changes in radiative forcing, and several studies have explored the interaction between the carbon cycle and the climate (Bouttes et al., 2012a; Ganopolski and Brovkin, 2017). Although fully coupling a carbon cycle model to a climate model is a more advanced endeavor, we are eager to explore this avenue in future research.

## 5 Conclusions

To understand the mechanisms of glacial–interglacial variability in the carbon cycle, this study examined the transient response

of the ocean carbon cycle to climate change, including the remarkable strengthening and weakening of the AMOC at the BA and YD transitions. This study represents an important step toward comprehensive transient simulations of the carbon cycle using an AOGCM, even though the changes in atmospheric $p$CO$_2$ are relatively small compared to those derived from ice core reconstructions. The importance of this study lies in its model–data comparisons of carbon isotope ratios that elucidate the impact of AMOC mode changes on the three-dimensional structure of water masses in the Atlantic, Southern, and Pacific

oceans.

Our model qualitatively simulates an increase in atmospheric $pCO_2$ during HS1. The calculated increase of approximately 10 ppm in atmospheric $pCO_2$ from 18 to 15 ka BP is mainly caused by increase in SST. The relatively modest increase in atmospheric $pCO_2$, compared to that of ice core records, might be attributable in part to relatively small increases in SST in the Southern Ocean. Additionally, comparison of carbon isotope signatures between the model and the data highlighted the scope for improvement in the representation of increased ventilation in the deep ocean and the North Pacific. Similarly, there is potential for improvement with respect to changes in surface biological productivity involving the nutrient cycle, including iron. Correction of these elements would substantially improve our understanding of the increase in atmospheric $pCO_2$ during the early deglaciation.

The drastic shifts in the AMOC during the BA and YD periods cause bipolar climate changes. These changes affect not only the temperature and salinity distributions but also the distributions of DIC and alkalinity. Interestingly, the cumulative effects of these changes on atmospheric $pCO_2$ appear to cancel each other out, resulting in only slight decrease during the BA period and increase during the YD period. It is noticeable that changes in $pCO_2^{os}$ due to variations in temperature and alkalinity play a major role in the reduction of atmospheric $pCO_2$ during the BA period.

To simulate transient changes in the carbon cycle, improvements in model accuracy, experimental configurations, and the models themselves are critical for capturing the dynamical and biogeochemical changes in the atmosphere and ocean. Further research is needed to identify the specific processes that influence changes in the ocean carbon cycle over different time scales in individual ocean basins. We emphasize the importance of analyzing carbon isotope variations that can provide valuable insights into past carbon cycle dynamics and contribute to comprehensive understanding of the glacial–interglacial variations in the ocean carbon cycle.

*Code and data availability.* The CCSR Ocean Component Model (COCO) is the ocean general circulation model of MIROC, and the code of COCO version 4.0 is included as part of MIROC-ES2L. The source code of MIROC-ES2L can be obtained from https://doi.org/10.5281/zenodo.3893386 (Ohgaito et al., 2021).

*Author contributions.* H.K. and A.O. designed the research. H.K. conducted the numerical experiments with help from T.O. H.K. performed the analysis and wrote the paper. A.O. and A.A.O. obtained funding and supervised the study. All authors discussed the results and commented on the manuscript.

*Competing interests.* The authors declare that they have no competing interests.

*Acknowledgements.* The authors express sincere gratitude to the two anonymous reviewers for their invaluable and constructive feedback on our manuscript. We would also like to thank Laurie Menviel for their insightful and helpful comments and for their expert editorial handling.

This work was supported by JSPS KAKENHI Grant Numbers JP17H06104, JP17H06323, JP19H01963, and JP21K13990. The ocean tracer model simulations in this study were performed at the Information Technology Center of the University of Tokyo. We thank James Buxton MSc, from Edanz (https://jp.edanz.com/ac), for editing a draft of this manuscript.

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

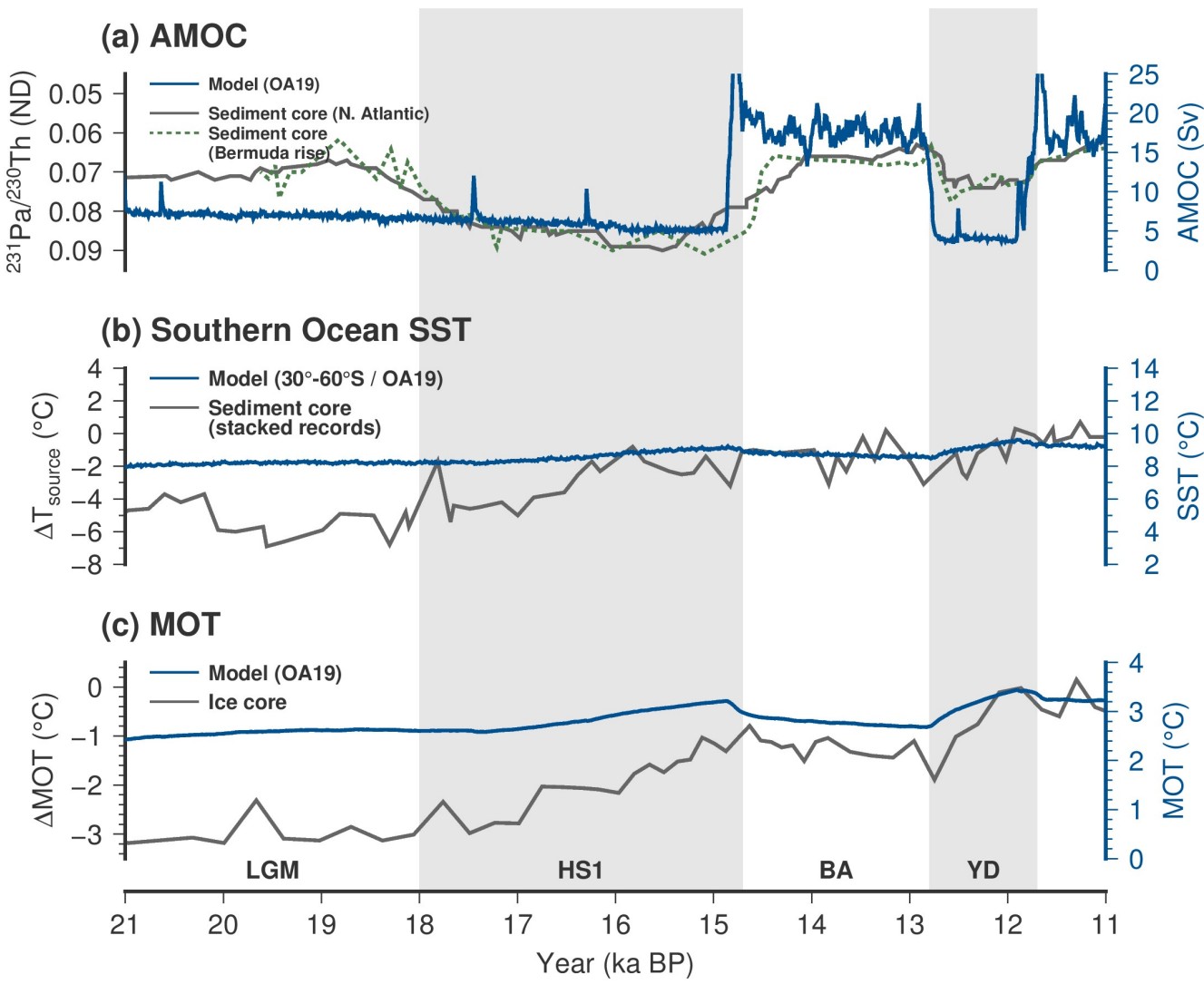

**Figure 1.** (a) Deglacial changes in the Atlantic Meridional Overturning Circulation (AMOC; Sverdrup) with $^{231}$Pa/$^{230}$Th reconstructed from Bermuda Rise sediment core data (McManus et al., 2004) and compiled sediment core data from the North Atlantic (Ng et al., 2018). The strength of the AMOC is defined as the maximum meridional volume transport between 30°N and 90°N at depths below 500 m. (b) Deglacial changes in sea surface temperature (SST) in the Southern Ocean (°C) and the difference in SST from the present day ($\Delta T_{source}$) (Uemura et al., 2018). (c) Deglacial changes in global mean ocean temperature (MOT; °C) and the difference in MOT from the present day ("Mix" of Bereiter et al. (2018)). The model output computed by the AOGCM (OA19) (Obase and Abe-Ouchi, 2019), shown by blue lines, is compared to reconstructions from geological data, shown by gray lines. The right axes relate to the model output and the left axes relate to the reconstructions.

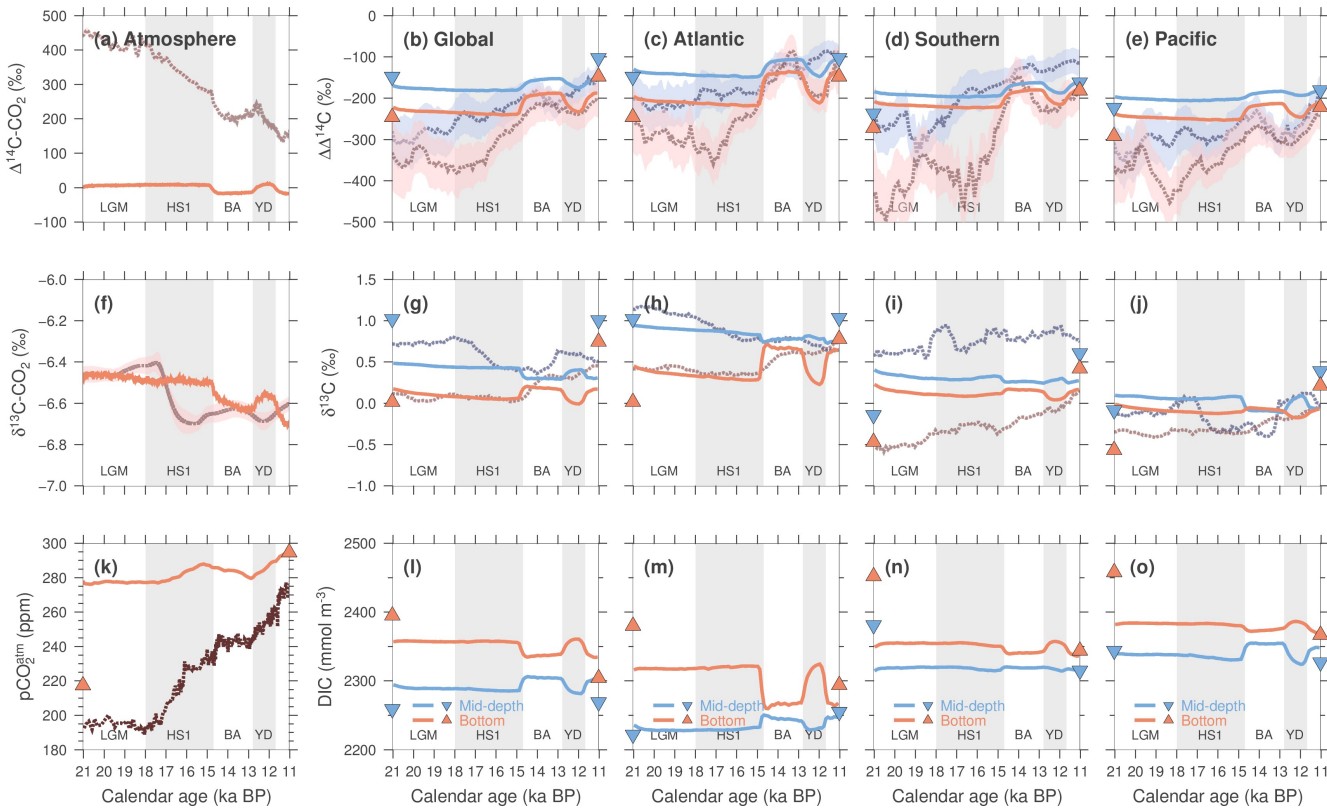

**Figure 2.** (a) Deglacial changes in $\Delta^{14}C$–$CO_2$ (‰) in the atmosphere (solid line) with the reconstruction of IntCal20 (dashed line; Reimer et al. (2020)). (b–e) Deglacial changes in $\Delta\Delta^{14}C$ (‰), difference in $\Delta^{14}C$ (‰) between the ocean and the atmosphere, averaged in the mid-depth (500–2000 m; blue solid lines) and deep global ocean (2000–5500 m; red solid lines) of the Atlantic Ocean ($40°S$–$90°N$), Pacific Ocean ($40°S$–$90°N$), and Southern Ocean ($90°$–$40°S$) with compiled sediment core data (dashed lines) of Rafter et al. (2022). (f) Deglacial changes in $\delta^{13}C$–$CO_2$ (‰) in the atmosphere (solid line) with the reconstruction (dashed line; Schmitt et al. (2012)). (g–i) Same as (b–e), respectively, except for $\delta^{13}C$ (‰) with compiled sediment core data of Muglia et al. (2023). (k) Deglacial changes in atmospheric $pCO_2$ (ppm; solid line) with ice core data (dashed line; Bauska et al. (2021)). (l–o) Same as (b–e), respectively, except for dissolved inorganic carbon (DIC; $\mathrm{mmol\,m^{-3}}$). LGM: Last Glacial Maximum, HS1: Heinrich Stadial 1, BA: Bølling–Allerød period, YD: Younger Dryas period. The triangles represent the values reported in Kobayashi et al. (2021), with the output of PI_sed plotted at the time of 11 ka BP and the output of LGM_all plotted at the time of 21 ka BP. PI_sed is an ocean carbon cycle model experiment conducted under preindustrial forcing, including carbonate sedimentation processes. LGM_all is an ocean carbon cycle model experiment conducted under LGM forcing, including enhanced salinity stratification in the Southern Ocean, iron fertilization from glaciogenic dust, and carbonate sedimentation processes.

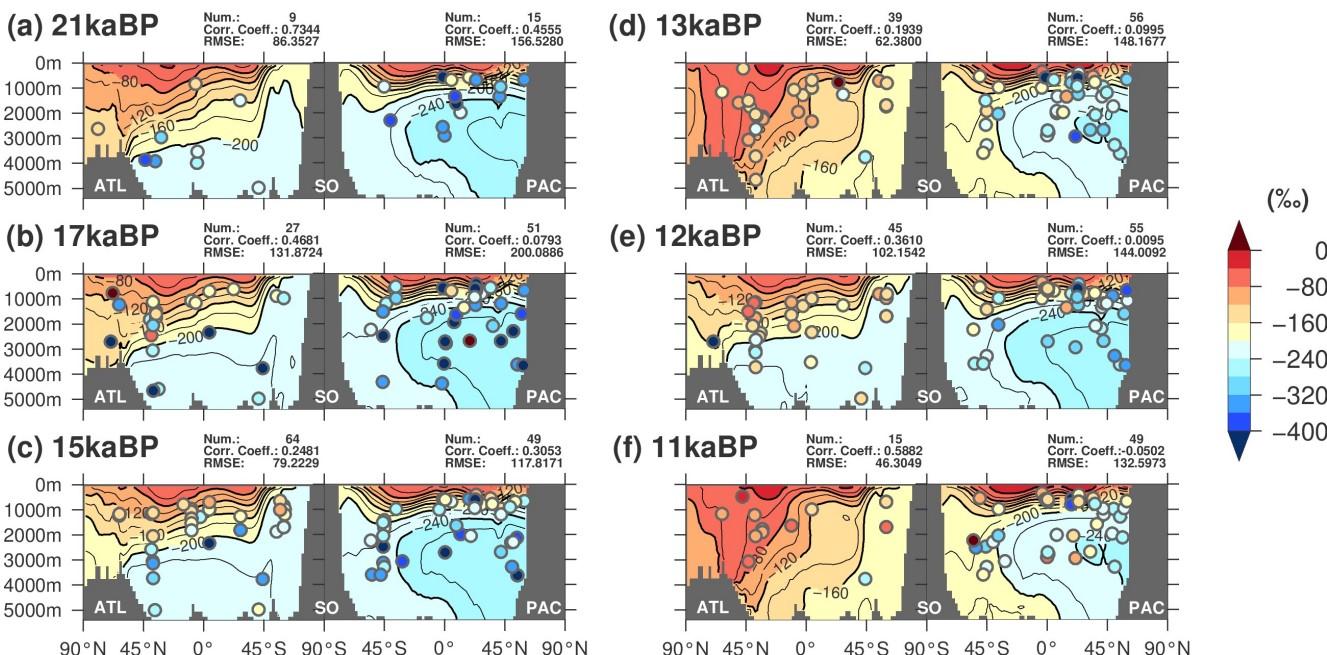

**Figure 3.** Oceanic zonal mean distribution of $\Delta\Delta^{14}$C (‰), which represents the difference in $\Delta^{14}$C between the ocean and the atmosphere, during key periods of the last deglaciation in the Atlantic and Pacific oceans. The specific periods of interest include (a) the Last Glacial Maximum (21 ka BP), (b) Heinrich Stadial 1 (17 ka BP), (c) just before the Bølling–Allerød (BA) transition (15 ka BP), (d) the BA warm period (13 ka BP), (e) the Younger Dryas period (12 ka BP), and (f) the Holocene (11 ka BP). The contour interval is 40‰. The sediment core records used in the figure are compiled in Rafter et al. (2022). Model results are averaged over 200 years, i.e., 100 years before and after each target year. However, for 21 ka BP, the average is taken from results spanning 21.0–20.9 ka BP; for 11 ka BP, the average is taken from results spanning 11.1–11.0 ka BP. The figure also includes a compilation of sediment core records where reconstructed values are plotted for 250 years before and after each target year. The vertical section represents all data within the relevant ocean basins. The abbreviations for the oceans are ATL for Atlantic Ocean, SO for Southern Ocean, and PAC for Pacific Ocean. The top-right notes indicate the number of data points, model–data correlation coefficients, and the root mean square error (RMSE) for both the Atlantic and the Pacific basins.

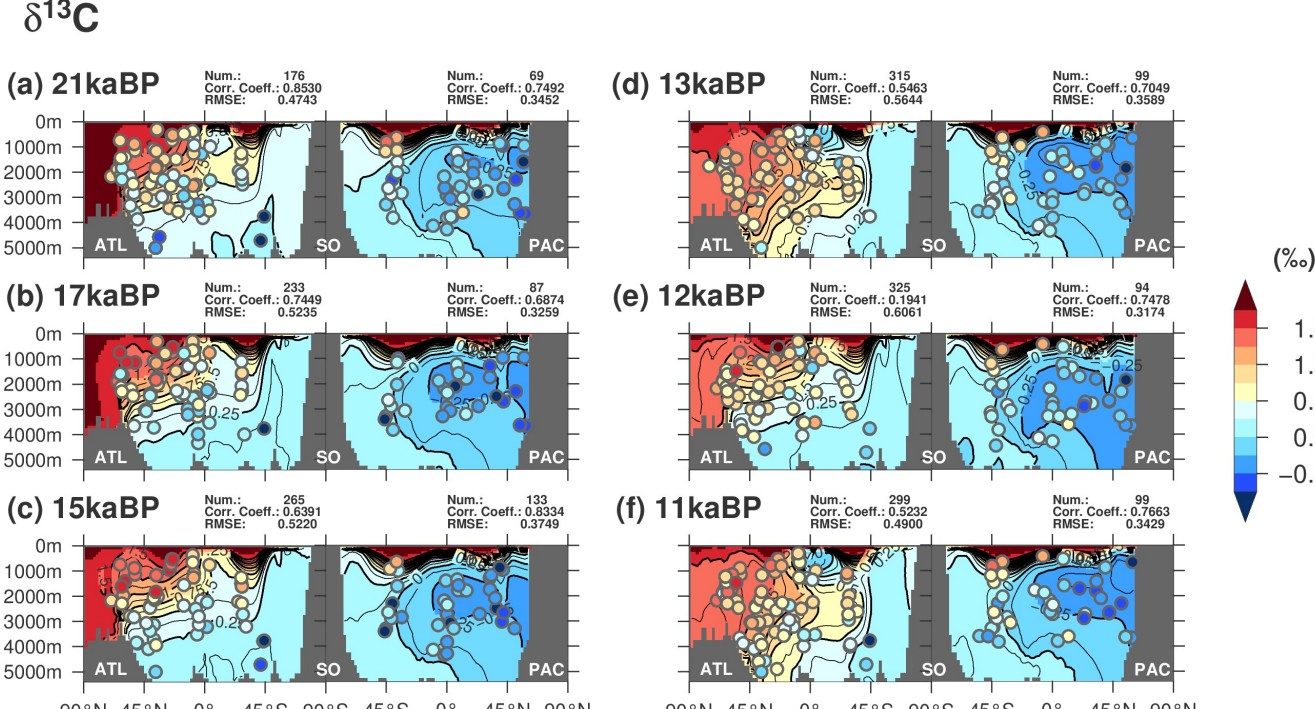

**Figure 4.** Oceanic zonal mean distribution of $\delta^{13}$C (‰) during the last deglaciation in the Atlantic and Pacific oceans. The contour interval is 0.25‰. The sediment core records used in the figure are compiled in Muglia et al. (2023). The period over which the model output is averaged is 100 years before and after the year of interest. The reconstructed values are also plotted for 100 years before and after the year of interest.

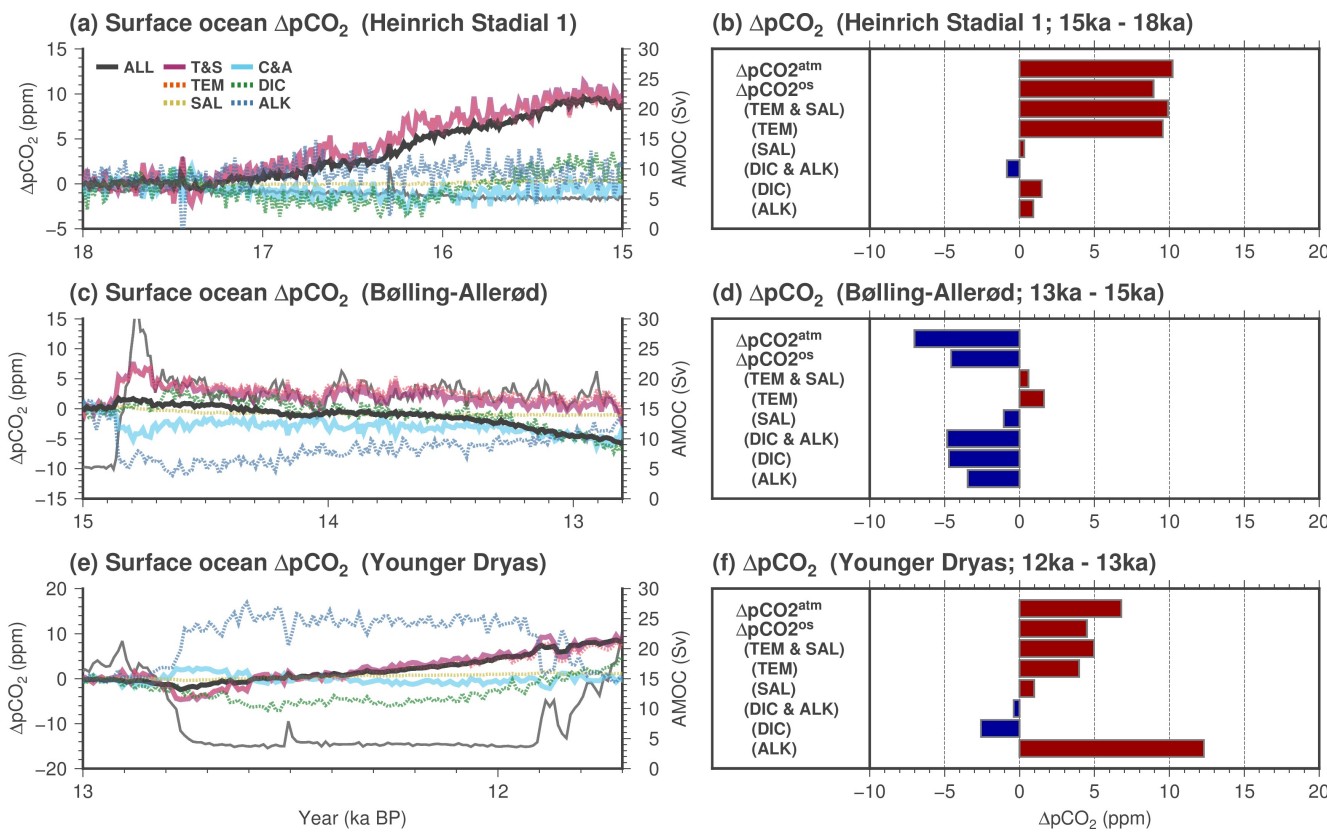

**Figure 5.** (a) Temporal changes in the partial pressure of sea surface $CO_2$ ($p$CO$_2^{os}$; ppm; gray) during Heinrich Stadial 1 (differences between 18 and 15 ka BP). The contributions of changes in temperature and salinity (purple), temperature (red), salinity (yellow), dissolved inorganic carbon (DIC) and alkalinity (cyan), DIC (green), and alkalinity (blue) to the changes in $p$CO$_2^{os}$ are shown. The thin gray line shows the time series of AMOC strength. (b) Temporal changes in the partial pressure of atmospheric $p$CO$_2$ and $p$CO$_2^{os}$ (ppm) during Heinrich Stadial 1. The contributions of changes in temperature and salinity, temperature, salinity, DIC and alkalinity, DIC, and alkalinity to the changes in $p$CO$_2^{os}$ are represented by different colored bars. (c) and (d) Similar to (a) and (b), respectively, but for the Bølling–Allerød period (differences between 15 and 13 ka BP). (e) and (f) Similar to (a) and (b), respectively, but for the Younger Dryas period (differences between 13 and 12 ka BP).

## Heinrich Stadial 1 (15ka - 18ka)

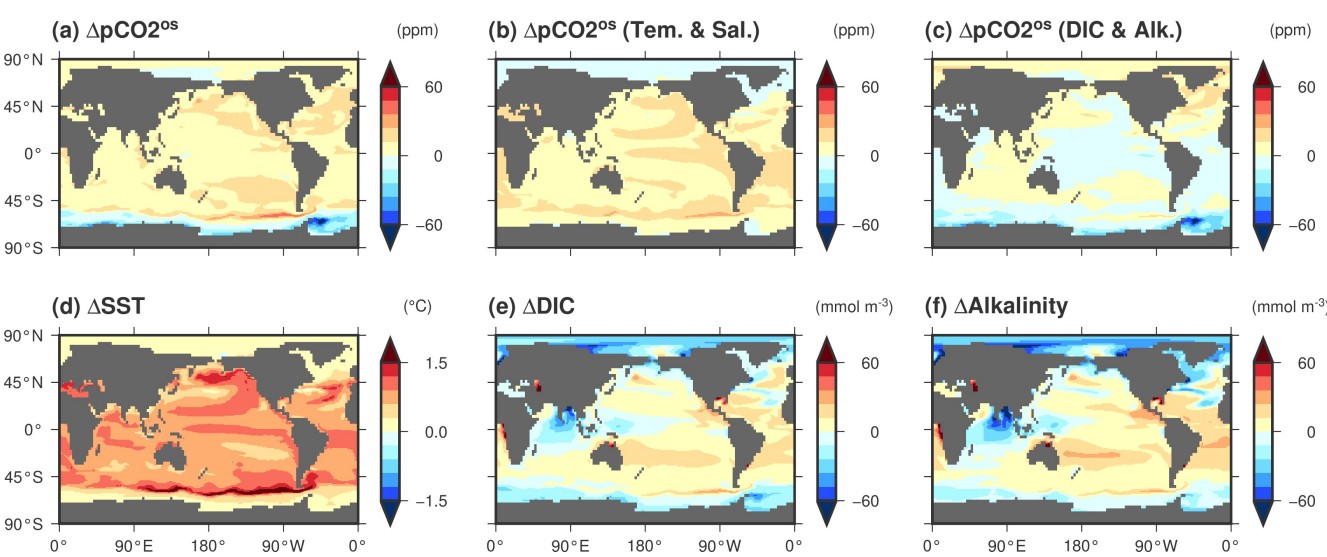

**Figure 6.** (a) Changes in partial pressure of sea surface $CO_2$ ($pCO_2{}^{os}$; ppm) between the early and late Heinrich Stadial 1 (differences between 15 and 18 ka BP). Changes in $pCO_2{}^{os}$ attributable solely to changes in (b) temperature and salinity and (c) dissolved inorganic carbon (DIC) and alkalinity, and to changes in sea surface (d) temperature, (e) DIC, and (f) alkalinity between the same periods.

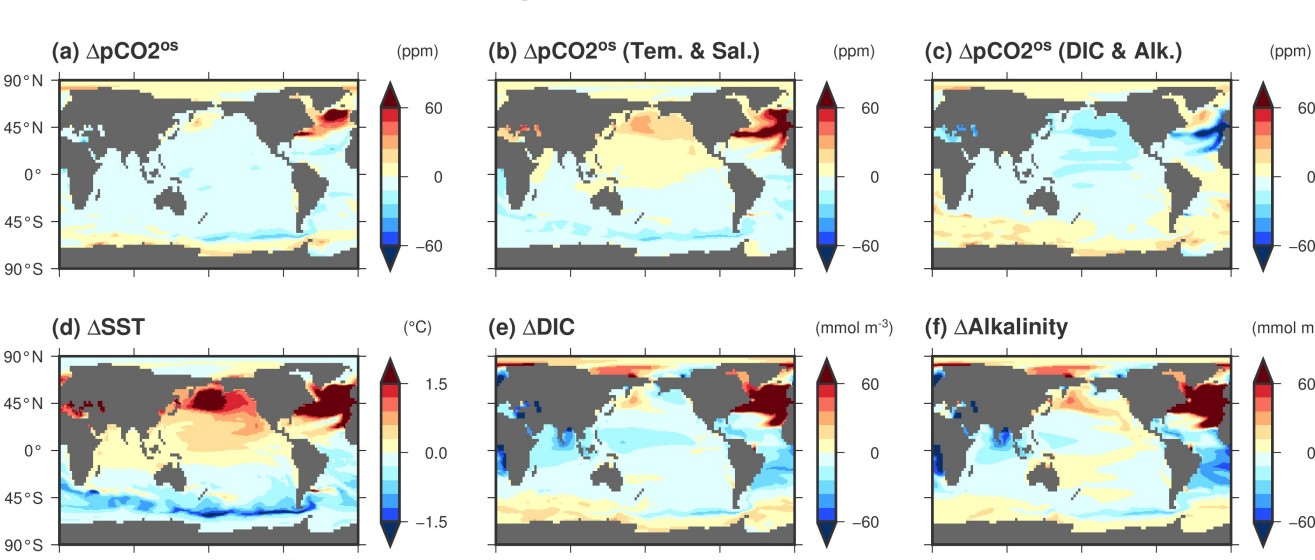

**Figure 7.** Same as Fig. 6 except for the Bølling–Allerød period (differences between 13 and 15 ka BP).

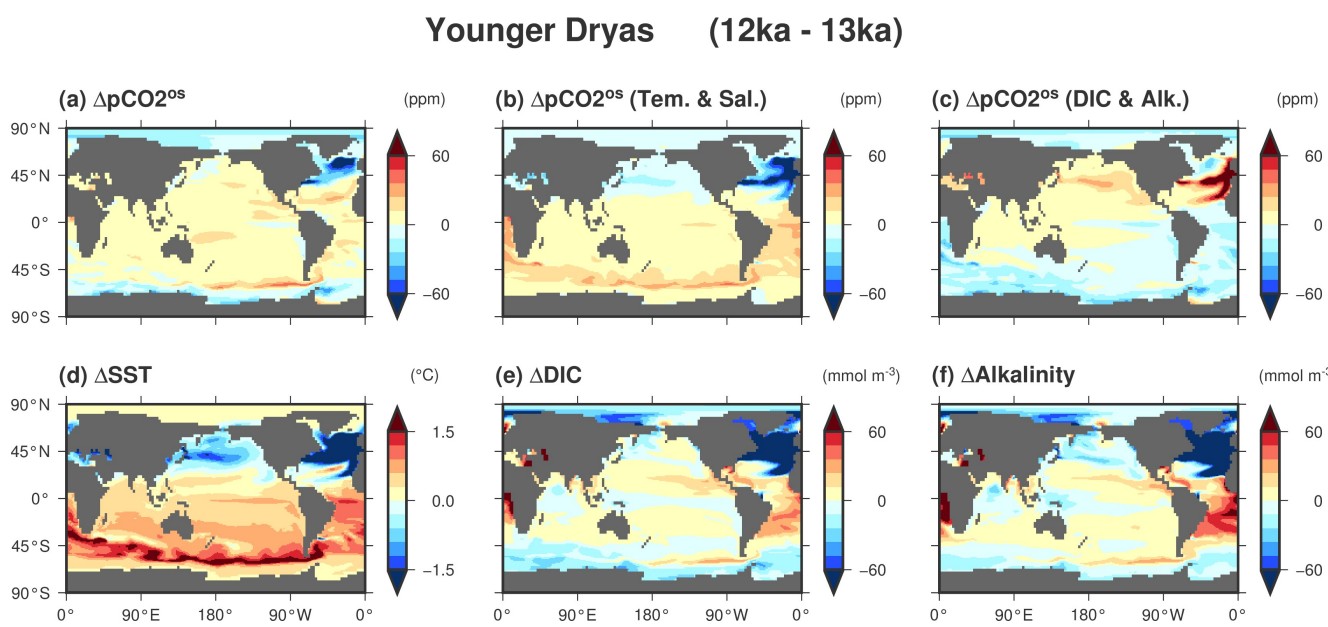

**Figure 8.** Same as Fig. 6 except for the Younger Dryas period (differences between 12 and 13 ka BP).