# Peer review of "Assessing transient changes in the ocean carbon cycle during the last deglaciation through carbon isotope modeling"

_EGUsphere, 2023_

## Referee Comment (RC2)

Review of "Assessing transient changes in the ocean carbon cycle during the last deglaciation through carbon isotope modeling" by Kobayashi et al.

The manuscript presents results of a transient simulation of the last deglaciation with an ocean carbon model forced with MIROC4m outputs (from Obase and Abe-Ouchi, 2019). The authors assess the simulated $pCO_2$, $\delta^{13}C$ and $\Delta^{14}C$ variations between 21 and 11 ka, in response to AMOC changes – in particular an abrupt increase at the onset of the Bølling-Allerød and Holocene and an abrupt decrease for the Younger Dryas. Using model-data comparison for both $\delta^{13}C$ and $\Delta^{14}C$ and a decomposition analysis of the $pCO_2$ changes, they are able to discuss to some extent the processes behind the large glacial-interglacial $CO_2$ variations recorded in ice cores, also building on Kobayashi et al. (2021) results. The observed model-data (mis)matches for $\delta^{13}C$ and $\Delta^{14}C$, which can differ, are informative in terms of processes and could pave the way for further modelling efforts targeting the last deglaciation.

I think that this study is well-suited for Climate of the Past and that the simulation, results, and analysis presented in the article are all worthy of publication. Overall, the manuscript is well-written, although there are places where the writing style and flow could be improved for the reader to follow more easily the scientific reasoning of the authors. To give this study more weight, I also think that the authors should not shy away from underlining more its strengths in a number of instances, explicitly connecting an important research question to a demonstrated knowledge gap ; as well as its weaknesses, hopefully providing for a clearer path forward for modelers interested in this question. I am providing below a number of points to help guide the authors in this direction. Since most of my comments are suggestions of improvement of the writing style/flow/clarifications, I recommend publication after minor revisions.

**General comments**

1. Abstract structure : The abstract starts out very abruptly with some of the methods (L1) and technical details (L2). I think that delineating the overall subject, why it matters, knowledge gaps and an explicit scientific question should come first for the reader to clearly understand the scientific reasoning behind the study. What is the scientific problem ? Why is it chosen ? Which methods are (thus) proposed to tackle it ? Some knowledge gap can be found right at the end of the abstract (L22-23), introduction (L47-51) or in the discussion (L273-277). Methodological elements are scattered in the abstract (L1-3, L7, L11, L19). I would recommend rearranging all of these elements so that the reader is not given the impression of a list of results, but of a logical approach to tackle an outstanding problem.

2. Introduction and study originality : Although the introduction has a clearer structure, I think that it lacks – like the abstract: (1) a clear explanation of the stakes (why is understanding these processes key?) ; (2) explicit connections between the limitations of previous studies (defining a knowledge gap), a scientific question, and the methods therefore chosen to tackle it. Without these elements, it is difficult for the reader to see the originality (asset, novelty) of this study.

3. Transitions : throughout the manuscript, there is a lack of transitions in-between paragraphs. This slows down reading as the reader has to stop and think about how the new idea is related to the previous one. There are parts (e.g. Discussion) where this absence of transition (and therefore of clear structure) makes the reader a bit lost. I would recommend using more link words (of other types of explicit connections) to make the reasoning more visible (and therefore easier to follow). I have underlined a few examples in the specific comments for guidance.

4. Limitations : Neither the consequences of using a fixed ocean volume (L123-125) and restoring term for carbon isotopes (L126-129) are discussed in Sect 4.3. Such a discussion would be welcomed. Indeed, Sect. 4.3 "Implications for future improvements to the model and experimental design" discusses in length potential model developments, but not much the improvements which could be made to the experimental design. However, both Snoll et al. (in review, 2023) and Bouttes et al. (2023) have demonstrated the influence of the choice of forcings (respectively, freshwater fluxes and interactive bathymetry) on model results. Perhaps describing potential improvements in terms of experiment design as well could help identify a clearer way forward (I mean some kind of shorter term strategy, as Sect. 4.3 and the conclusion are both sending off a rather vague "we need to improve models" message – which is relevant, but in large part a long term endeavor).

5. Tense : I think that the past (e.g. L3, L8, L9… and throughout the manuscript) and past perfect (e.g. L20, L22…) tense tend to make statements less effective than present tense.

**Specific comments**

L5 and L7 : "increased", "decreasing trend". By how much? More frequent quantification would be welcomed.

L12-13 : "We found that…". This statement seems to be contradictory with Fig. 2, unless "after the onset of the BA" is specified.

L17-18 : "smaller atmospheric $pCO_2$ changes than ice core data". Are you referring to changes during HS1 or during the whole deglaciation?

L26-27 : the "which is" proposition interrupts the "from … to…" statement, giving a jerky rhythm to the sentence.

L31 : An example of where a transition (e.g. "To decipher the reasons behind those changes,…") would be welcomed. Same for L34 (e.g. "In particular,…", with no line jump).

L37 : "Therefore" provides for an incomplete argument for different $\delta^{13}C$ in water masses, for only the impact of fractionation during photosynthesis is described before – not including the impact of ventilation on the isotopic signal.

L54 : Are you using the plural form to designate both the soft tissue and the carbonate pumps combined? According to Kohfeld and Rigwell (2009), strictly speaking, the biological pump (singular) encompasses both.

L55-59 : I find the explanation of the limitations of the use of steady-state differences (and therefore, of the asset of using transient simulations) to be incomplete and therefore only partially convincing. Could you elaborate ?

L59 : The change of tense ("will improve" to "have been conducted") makes it confusing for the reader to understand what exactly is the knowledge gap, and what is new in this study with respect to previous studies.

L66 and L71 : the lack of transition with "other related studies", "several related studies" makes it difficult to follow the reasoning and understand where this paragraph is going. I would recommend connecting ideas rather than juxtaposing them.

L71-75 : this sentence contains many processes and seem therefore very long. Could it be divided into two? As for the references L75-76, it is unclear whether they refer to either one, or all, of the mentioned processes.

L78-80 : it seems unclear to me (1) what were the conclusions of those previous studies, (1) what is their limitation you are mentioning, and therefore (3) the novelty brought by your study. Could you elaborate ?

L90 : "boundary conditions" is a rather technical jargon which, depending on the model setup, can designate various things. I would prefer using "was forced with" instead.

Sect. 2.2 : A little surprisingly, the PMIP4 protocol (Ivanovic et al., 2016) is not mentioned. It could be worth noting whether the AOGCM simulation by OA19 followed the PMIP4 protocol for deglacial simulations, and if not, where it differed.

L106 : Without reading the quoted paper, these values and how they were chosen seem a little mysterious.

L115 : Please give out here a quantified value of the $pCO_2$ simulated in the 2021 paper. Same in L162. The authors could also consider adding a triangle for this value in Fig. 2k.

L124 : Ocean volume change would also induce changes in alkalinity and nutrients, not only dissolved matter concentration.

L142 : "underestimates". As the simulated values are less negative than the reconstructed ones, I am unsure whether "underestimates (how low these values are)" or "overestimates" should be used. This English vocabulary should be checked.

L155 : "approximately 80 ppm". The value here is lower than the one chosen L28. It is best to use consistent numbers.

L156 and L161 : "One possible explanation" / "Another possible explanation". This vocabulary may give the reader the impression that theses reasons are mutually exclusive, which is not the case.

L172 : "at times". Could you specify which times? and why?

L175-178 : This sentence brings very little new elements with respect to L174-175 (except the numerical value of -200 permil).

L180-181 : The link between the AMOC and the deep Pacific ventilation is unclear.

L183 : "do not provide clear indication of the intrusion of young water masses". Where is this evidence? Please quote a figure number.

L184-185 : "whereas the model experiment does not show such pronounce change [during HS1]". I find this statement to be slightly misleading, for I would expect that the absence of a large increase of simulated $\Delta^{14}C$ to be related not to a model error during HS1, but to the very high values (wrt. reconstruction) inherited from the initial state (i.e. not enough carbon sequestration in the ocean at the LGM).

L188 : " $\Delta^{14}C$-CO2 (Fig. 2a)". Please explain this choice of plot.

L190-195 : This paragraph feels disconnected from the previous descriptions. A transition would probably help integrate it explicitly in the reasoning.

L200 : Could we perhaps see a figure of biological production as well in Supplementary?

L199-201 : How is a reduced vertically gradient in response to a stronger AMOC related to the sensitivity of $\delta^{13}C$ to climate change?

L204-206 : The reasoning seems incomplete. How should we interpret this different model-data agreement for the North Atlantic / Southern Ocean?

L207: "the AMOC is intensified". Add "and deepens".

Sect 3.3.2 and 3.3.3 : Like in Sect 3.3.1, I would welcome here a quantification of the atmospheric pCO$_2$ changes occurring during the BA and YD.

L260 : "the AMOC resumes over time". This feels like an inaccurate description of the AMOC variations, as Fig. 1 rather shows a stabilization during the BA after an overshoot at the onset of the BA.

L280 : "after the BA transition are generally consistent". Starting out this paragraph like this is a bit surprising (in terms of chronology), as this statement does not acknowledge the large model-data gap before the BA transition.

L193 : "a longer period". Please quantify the difference. The forcing in terms of meltwater fluxes also seem to differ between the two studies.

L297-300 : Although this seems like conclusive remarks, I am not coming out of the paragraph with a clear idea of what the comparison to the Pöppelmeier et al. (2023) results actually brought to the table. Clarifying the transition (L286, L301) could help.

L303 : "to the deep ocean". Do you mean deep Atlantic ocean? This mismatch seems larger in the Atlantic than Southern Ocean.

L312-313 : Could you specify? As such, this is an underwhelming statement.

L321 : "AMOC weakening" could be changed to "AMOC weak state", since the actual weakening occurs at the onset of these events.

L323 : Shouldn't the contribution of SST during YD and HS1 also be mentioned?

L325 : Could we perhaps see a figure with the vertical gradients in Supplementary?

L327 : "increased SST and reduced surface ocean alkalinity". During which part(s) of the deglaciation specifically? Or do you mean the whole deglaciation?

L329 : The absence of transition is more notable as we go back in time (reversed chronology here).

L332 : "In contrast" seems like the wrong link word here.

L350-351 : "AOGCM of MIROC", "EMIC of iLOVECLIM". The model classification is irrelevant here. The difference in model resolution could be mentioned if the authors would like to propose it as a potential cause for the observed model difference (although difference in forcings could also play a role).

L420 : Add something like "at the BA and YD transition respectively"

L422c : "relatively modest". Please quantify again.

Fig. 1 : (a) Please specify in the legend what the metric used for the AMOC strength (max at 26°N?). Also, you could consider adding triangles for the Kobayashi et al. (2021) values as in Fig. 2.

Fig. 2 : The "PI_sed" and "LGM_all" simulations could be (briefly) described in the legend so that the reader knows in which way they differ from the transient run initial state without having to look for the simulation description in the 2021 paper.

Fig. 3 : Please specify which ocean basin is where on the plots (Atl = left, Pac = right).

Fig. 4 : The colorbar seems to saturate for very negative $\delta^{13}$C values in marine core data. Also, the contour interval is too dark and narrow to see to subsurface values. A few adjustments (and larger plots) could improve visibility.

Fig. 3 and 4 : Please consider calculating the RMSE for both proxies and all periods.

Fig. 5 : The thin gray line is not very visible on all panels. I would suggest finding an adjustment (e.g. lowering opacity for superimposed curves) to improve visibility.

**Technical comments**

L2 : "the effects" on what ? Please specify.

L6 : Introduce "(BA)" abbreviation here.

L10 : "Meanwhile" or "in the meantime"

L26 : "has transitioned" → "transitioned"

L43 : typo "infer"

L52 : "those" → "the"

L119 : "ocean biogeochemical cycle" → add "carbon"

L230 : "of" → "the"

L234 : "those" → "the"

L292 : "almost completely" → drastically (or synonyms)

L317 : Add "… to several factors. First, …" ; L319 : "Moreover" → "Second"

L359 : "are synchronized"

L390 : typo "lowering"

L397 : plural "developments"

L436 : "what" → "which"

Most figures, especially Fig. 3 and 4, would gain to be larger.

---

## Author Comment (AC1)

**Response to Reviewer 1**

The authors are grateful to the reviewer for their valuable input. The recommendations of the reviewer have been carefully incorporated into the revised manuscript, as described in the following (text colored blue is extracted from the reviewer's feedback; text in italic font represents excerpts from the revised manuscript). Additionally, the guidance that we received regarding figure enhancements was extremely helpful, and we have made the adjustments necessary to improve both the clarity and the impact of the figures.

Kobayashi and colleagues present a novel offline simulation conducted with a carbon isotope-enabled biogeochemical model, forced with climate data from a transient MIROC4m simulation of the last deglaciation. Their investigation focuses on the relationship between changes in atmospheric $CO_2$, stable and radiocarbon, and the varying AMOC during that period. The simulation reveals relatively minor changes in atmospheric $CO_2$ concentration compared to ice core reconstructions, while demonstrating that changes in water mass ventilation and sourcing align with proxy reconstructions. The authors further analyze the simulated changes in p $CO_2$, attributing them to physical (temperature and salinity) and biogeochemical (dissolved inorganic carbon and alkalinity) drivers, revealing complex interactions and compensating effects.

The manuscript is well-written and well-illustrated. While similar studies have been previously conducted with intermediate complexity models, this study represents a significant step towards comprehensive transient simulations with an AOGCM, despite not fully achieving this here. The authors transparently acknowledge certain critical factors during the last deglaciation, such as sea-level rise, ice sheets, and Southern Ocean sea surface temperature biases, which were not accounted for in this study. Considering all these processes for a deglacial simulation is very

challenging in such a complex model, making it understandable that they are not fully considered here. Therefore, I recommend the manuscript for publication in Climate of the Past after minor revisions. Detailed comments outlining specific areas for improvement are provided below.

Thank you for your understanding of the key aspects of our research. We validated the calculated carbon isotope ratios by comparing them with sediment core records, and we assessed the current advances and limitations regarding simulation of the deglacial changes in the carbon cycle.

**Comments:**

L4: Better introduce the abbreviation AMOC already here than in L7.

As recommended, we have defined the meaning of the acronym "AMOC" at the point at which it is first used. In response to feedback from reviewer #2, we have completely revised the structure of the abstract.

L5: Here and in the following, I would replace "atmospheric partial pressure of carbon dioxide" just with "atmospheric concentration of $CO_2$".

We agree with this suggestion. The term *"atmospheric partial pressure of carbon dioxide"* has been replaced with *"atmospheric concentration of carbon dioxide ($pCO_2$)"*.

L30: deglacial period not deglaciation period.

The term *"deglaciation period"* has been replaced with *"deglacial period."*

L43: investigate not inter.

The typographical error *"inter"* has been corrected to *"infer."*

L80: What is meant by vertical one-dimensional distribution in this context? Previous studies have evaluated the 3D distribution of data.

As you correctly mentioned, previous EMIC experiments calculated the three-dimensional distribution of carbon isotope ratios. However, in the context of model–data comparisons, the focus often remains on their horizontally averaged one-dimensional distribution over ocean basins. This paragraph has been carefully revised to present the information clearly.

L87-92: The setup of the model is not fully clear to me. In line 87 it is mentioned that a BGC model was coupled to an ocean model, but in line 90 it is stated that the BGC model was forced with MIROC output. Does this mean that the coupled BGC-ocean model was forced with atmospheric boundary conditions of MIROC? Later on it reads to me as if the BGC model is run entirely in offline mode. Can the authors clarify this in section 2.1?

We conducted offline experiments of the ocean biogeochemical cycle, forced with the output from the AOGCM MIROC. The ocean biogeochemical cycle model was run within the COCO ocean model framework. We have revised Section 2.1 to explain the experimental design more clearly.

L91: Can you briefly mention how the MIROC transient simulation was forced, e.g., freshwater fluxes GHG concentrations etc.?

The MIROC 4m experiment focusing on the last deglaciation was performed according to the PMIP protocol (Ivanovic et al., 2016, GMD) with respect to the changes in orbital parameters and greenhouse gases throughout this period (Obase and Abe-Ouchi, 2019). In that earlier work, they fixed the ice sheet to the 21 ka BP state of the ICE-5G reconstruction. Freshwater inputs from the Northern Hemisphere ice sheets deviated from the PMIP protocol after the latter half of Heinrich Stadial 1, but we never stopped the freshwater input at any time (details in Obase and Abe-Ouchi). This procedure was intended to synchronize the simulated AMOC variations with

those reconstructed from sediment core records and the associated climatic changes that occurred during the Bølling–Allerød and Younger Dryas periods.

The information on the MIROC 4m experimental design is explained in Section 2.2 of the revised manuscript.

L110: Why were pre-industrial and not LGM values used to initialize these atmospheric values?

As you noted, atmospheric $CO_2$, $\delta^{13}C$, and $\Delta\Delta^{14}C$ values were all initialized to preindustrial levels for the spin-up under the 21 ka BP forcing. This is consistent with the method used by Kobayashi et al. (2021, *Sci. Adv.*; K21). This approach might increase discrepancies between the model and the observational data, especially during the early deglaciation. However, even if the spin-up occurs using LGM atmospheric values, the discrepancy between the model and the data would become larger in the latter half of the deglaciation.

L138: Can you give numbers for the AMOC strengths during the LGM and the Holocene?

The strength of the AMOC is defined as the maximum meridional volume transport between 30°N and 90°N at depths below 500 m. The calculated strengths of the AMOC at 21 and 11 ka BP are 9.0 and 17.4 Sv, respectively. The derivation of these values has been described in the revised manuscript. Additionally, the definition of the AMOC has been added to the footnote of Fig. 1.

L146: According to Fig. 2j the Pacific appears to me simulated in rather good agreement with the data. Certainly, much better than the Southern Ocean.

As you mentioned, the calculated changes are closer to the data in the Pacific Ocean than in the Southern Ocean. The text has been revised to focus on the Southern Ocean.

L151: Maybe it is worth mentioning, that indeed the agreement of Kobayashi2021 is much better in the deep Southern Ocean, but quite a bit worse for the mid-depths. It therefore appears to me that it is not as simple as including these processes.

We agree with your comment. The reproduction of carbon isotopes of the Southern Ocean in K21 is notably more accurate than in this study; however, changes appear somewhat exaggerated in other regions, e.g., the Pacific Ocean. It is currently difficult to achieve a consistent scenario that explains all the changes in the global ocean. We have added a description in the revised manuscript outlining this difficulty:

*"A comparison between the two studies highlights the advances made by Kobayashi et al. (2021) in capturing the dynamics of the Southern Ocean, suggesting that incorporating the processes considered in their research could improve model-data agreement. However, the challenge remains that their LGM simulation slightly overestimates changes in the glacial Pacific. These discrepancies highlight the difficulty of achieving consistent scenarios that account for all changes in the global ocean within a model."*

L160: Can you give a reason why the SST difference between the LGM and Holocene is so small compared to observations?

In the MIROC LGM experiment, the AMOC oscillates with a very long period. The initial state of this study corresponds to the physical field long after the AMOC has weakened. Consequently, the meridional heat transport weakens, and both the deep sea and the Southern Ocean tend to warm. Furthermore, although the climate sensitivity of the MIROC model is not small at 3.9°C, the LGM SST in the Southern Ocean tends to be low (Obase

et al., 2023, Clim. Past. discuss). With respect to this factor, Obase et al. (2023, CPD) discussed how the weak LGM AMOC in the MIROC, as well as ice sheets and cloud radiation, influence the LGM SST.

MIROC exhibits small SST differences between the LGM and the Holocene, particularly in the Southern Ocean. In fact, a recent six-model intercomparison of the last deglaciation shows that MIROC tends to have smaller SST changes than other models, although the equilibrium climate sensitivity of the MIROC model is not small at 3.9°C (Fig. S4 of Obase et al., 2023, *Clim. Past. Discuss*). One possible explanation is the state of the AMOC during the LGM. The initial state in this study is derived from the physical field long after the AMOC has weakened. It is important to note that the AMOC oscillates on millennial timescales in the MIROC under the LGM condition. Consequently, the weakened meridional heat transport tends to warm the Southern Ocean more than in other models. Furthermore, Obase et al. (2023, *Clim. Past. Discuss*) discussed that the asymmetric responses to warming and cooling, associated with LGM ice sheets and cloud radiation, might contribute to the smaller LGM SST changes despite the relatively high climate sensitivity.

L178: There appears to be very little change during the transition from the LGM to HS1, which makes sense, since the AMOC also remains virtually constant. Maybe this needs to be hence slightly rephrased.

As you correctly noted, weakening of the AMOC during Heinrich Stadial 1, as suggested by $^{231}Pa/^{230}Th$, is very small in this study. This statement has been changed to refer to the period after the Bølling–Allerød transition.

L203: As mentioned before, the difference between the LGM and HS1 seems very small in the simulation, because the AMOC also changes very little. In contrast, a larger decrease is observed in the data.

As you mentioned, the changes in $\delta^{13}C$ during Heinrich Stadial 1 are

minimal compared to the data, which can be attributed to the small changes in the AMOC in the MIROC. The following sentences have been added in the revised manuscript:

*"However, the observed $\delta^{13}C$ change is relatively small compared to the sediment core data because the AMOC change is less pronounced than expected from the $^{231}Pa/^{230}Th$ reconstruction (McManus et al., 2004; Ng et al., 2018)."*

L209: Mention that this is most pronounced in the Atlantic.

In accordance with your suggestion, the following sentence has been added in the revised manuscript: *"This change is most pronounced in the Atlantic."*

L210: It is important to note, that two different things are compared here. The model output has an annual resolution and therefore shows the "true" perturbation magnitude. On the other hand, the reconstructions from marine sediments are smoothed out by processes such as bioturbation, coring artifacts, etc. It is therefore expected that the signal amplitude is bigger in the model than the data for such short perturbations like the YD.

We appreciate your insightful advice on the comparison between the model output and the sediment core data. We have included this note in the Discussion section regarding the explanation of the differences in temporal resolution between the model data and the sediment core data.

L211: Maybe explicitly mention that this is for the reconstructions.

This comment appears to refer to the point regarding L210. Building on the previous response, we have included discussion of the inaccuracies in sediment dating and temporal resolution issues associated with the sediment core data.

L224: This is rather surprising to me, as the AMOC strength actually doesn't change much, but the change in carbon export is rather large in the South Pacific. I'm therefore wondering whether this can really be attributed to an AMOC weakening, or whether other processes dominate this effect? From the pattern and the fact that this negative anomaly persists throughout the deglaciation independent of the AMOC strength, suggest to me that this is primarily a signal of increased iron limitation, which is mentioned in the text.

During Heinrich Stadial 1, the AMOC weakens slightly, resulting in nutrient accumulation in the lower cells of the meridional overturning circulation and in reduced nutrient supply to the South Pacific gyre. Furthermore, we hypothesize that increased iron limitation also contributes to the reduction in carbon export in the South Pacific. For more quantitative understanding, comparison with sensitivity experiments that fix the atmospheric iron supply at the LGM state is necessary. Additionally, multi-model intercomparisons using different iron cycle models and ocean models would be valuable to confirm the validity of the proposed mechanisms.

L227: By which mechanism propagate these changes to the North Pacific?

When the AMOC is strengthened during the Bølling–Allerød period, more nutrients are redistributed to the upper cells of the AMOC. These increased nutrients are outcropped in the lower latitudes of the Southern Ocean. Consequently, the transport of nutrients by surface and intermediate waters from the Southern Ocean to the North Pacific increases.

L304-310: It could also be that the AMOC weakening is too strong in the model.

Comparison of the model and the sediment data for $\Delta\Delta^{14}C$ and $\delta^{13}C$ supports the suggestion that weakening of the AMOC during the Younger Dryas period might be overly pronounced in the model. The relevant

sentences have been amended in the revised manuscript as follows:

*"Another important factor is the weakening of the simulated AMOC during the YD. The comparison of the model and sediment data for $\Delta\Delta^{14}C$ and $\delta^{13}C$ suggests that the weakening of the AMOC during the YD may be overly pronounced in the model."*

L334-344: The terrestrial biosphere plays an important role for atmospheric $\delta$13C, which I think should be mentioned here as well (see e.g., Jeltsch-Thoemmes et al., 2019, doi:10.5194/cp-15-849-2019).

As you correctly noted, changes in vegetation also contribute to the deglacial changes in atmospheric $\delta^{13}C$–$CO_2$. We have added the reference that you suggested, reorganized the text in the Discussion section, and summarized the description of terrestrial carbon reservoirs in Section 4.3.

Fig. 1: I understand the intention to make the model and data timeseries overlap for better comparability. However, I find this in panel c somewhat misleading, as there is a factor of more than two between both y-axes. I would like to see the same increment for both axes as it is done in panel b. Further, can a panel showing global mean surface temperature be added, for instance compared to the data assimilation by Osman et al., 2021 (doi:10.1038/s41586-021-03984-4)?

Thank you for your feedback. Panels (b) and (c) in Fig. 1 have been adjusted to have the same scale for the vertical axis.

Additionally, the global mean surface temperature (GMST) changes are shown below. Because the ice sheet is fixed at the state of the LGM, the GMST has increased only by approximately 2.5°C.

[Figure]

Fig. 3 and 4: Can you add a similar figure like these two, but for DIC (anomaly?), to better illustrate where the carbon was stored in the course of the deglaciation?

A figure showing the temporal changes in DIC was added to the Supplementary Figures.

Figs. S4 and S5: The panels are very small and hence hard to read. Can they be made larger, splitting the panels into different rows or columns?

The panels in Figs. S4 and S5 have been enlarged and reorganized into different rows to improve readability based on the feedback received.

Fig. S7 to S9. I find both these figures very instructive and if possible would like to see them in the main text.

Figures S7–S9 have been incorporated into the main text as Figs. 6–8, respectively.

---

## Author Comment (AC2)

**Response to Reviewer 2**

The authors greatly appreciate the insightful feedback from the reviewer. We have incorporated all the recommended changes detailed below (comments in blue are extracted from the reviewer's feedback; text in italic font represents excerpts from the revised manuscript). We also appreciate the helpful comments regarding the structure of the manuscript, which we have amended to improve coherence between sentences. Additionally, the comments regarding the figures were highly informative, and we have taken the necessary steps to improve their clarity.

The manuscript presents results of a transient simulation of the last deglaciation with an ocean carbon model forced with MIROC4m outputs (from Obase and Abe-Ouchi, 2019). The authors assess the simulated $pCO_2$, $\delta^{13}C$ and $\Delta^{14}C$ variations between 21 and 11 ka, in response to AMOC changes – in particular an abrupt increase at the onset of the Bølling-Allerød and Holocene and an abrupt decrease for the Younger Dryas. Using model-data comparison for both $\delta^{13}C$ and $\Delta^{14}C$ and a decomposition analysis of the $pCO_2$ changes, they are able to discuss to some extent the processes behind the large glacial-interglacial $CO_2$ variations recorded in ice cores, also building on Kobayashi et al. (2021) results. The observed model-data (mis)matches for $\delta^{13}C$ and $\Delta^{14}C$, which can differ, are informative in terms of processes and could pave the way for further modelling efforts targeting the last deglaciation.

Thank you for summarizing the main aspects of our manuscript. We appreciate your encouragement regarding our transient simulation during the last deglaciation and your recognition of the insights gained from the model–data comparisons.

I think that this study is well-suited for Climate of the Past and that the simulation, results, and analysis presented in the article are all worthy of

publication. Overall, the manuscript is well-written, although there are places where the writing style and flow could be improved for the reader to follow more easily the scientific reasoning of the authors. To give this study more weight, I also think that the authors should not shy away from underlining more its strengths in a number of instances, explicitly connecting an important research question to a demonstrated knowledge gap; as well as its weaknesses, hopefully providing for a clearer path forward for modelers interested in this question. I am providing below a number of points to help guide the authors in this direction. Since most of my comments are suggestions of improvement of the writing style/flow/clarifications, I recommend publication after minor revisions.

Thank you for your thoughtful review and positive assessment of the suitability of our study for publication in Climate of the Past. We appreciate your feedback on improving the readability and scientific integrity of the manuscript. We have carefully considered your suggestions to strengthen the links between the research questions and identified knowledge gaps. Our revision focused on improving the writing style and addressing weaknesses to ensure clearer comprehension by other modelers interested in this subject area. Your guidance was instrumental in helping us refine our work, and we have incorporated your suggestions to improve the manuscript prior to its submission.

**General comments:**

1. Abstract structure: The abstract starts out very abruptly with some of the methods (L1) and technical details (L2). I think that delineating the overall subject, why it matters, knowledge gaps and an explicit scientific question should come first for the reader to clearly understand the scientific reasoning behind the study. What is the scientific problem? Why is it chosen? Which methods are (thus) proposed to tackle it? Some knowledge gap can be found right at the end of the abstract (L22-23), introduction

(L47-51) or in the discussion (L273-277). Methodological elements are scattered in the abstract (L1-3, L7, L11, L19). I would recommend rearranging all of these elements so that the reader is not given the impression of a list of results, but of a logical approach to tackle an outstanding problem.

We appreciate this valuable feedback regarding the structure of the abstract. The abstract in the revised manuscript has been completely restructured based on your advice.

2. Introduction and study originality: Although the introduction has a clearer structure, I think that it lacks – like the abstract: (1) a clear explanation of the stakes (why is understanding these processes key?); (2) explicit connections between the limitations of previous studies (defining a knowledge gap), a scientific question, and the methods therefore chosen to tackle it. Without these elements, it is difficult for the reader to see the originality (asset, novelty) of this study.

We have incorporated this feedback and completely revised the Introduction section. We paid particular attention to the transitions and reorganized the text to create better connections between paragraphs.

3. Transitions: throughout the manuscript, there is a lack of transitions in-between paragraphs. This slows down reading as the reader has to stop and think about how the new idea is related to the previous one. There are parts (e.g. Discussion) where this absence of transition (and therefore of clear structure) makes the reader a bit lost. I would recommend using more link words (of other types of explicit connections) to make the reasoning more visible (and therefore easier to follow). I have underlined a few examples in the specific comments for guidance.

We appreciate your valuable advice. We have revised the text with particular attention to transitions and we noted these points in the English

proofreading submission.

4. Limitations: Neither the consequences of using a fixed ocean volume (L123-125) and restoring term for carbon isotopes (L126-129) are discussed in Sect 4.3. Such a discussion would be welcomed. Indeed, Sect. 4.3 "Implications for future improvements to the model and experimental design" discusses in length potential model developments, but not much the improvements which could be made to the experimental design. However, both Snoll et al. (in review, 2023) and Bouttes et al. (2023) have demonstrated the influence of the choice of forcings (respectively, freshwater fluxes and interactive bathymetry) on model results. Perhaps describing potential improvements in terms of experiment design as well could help identify a clearer way forward (I mean some kind of shorter term strategy, as Sect. 4.3 and the conclusion are both sending off a rather vague "we need to improve models" message – which is relevant, but in large part a long term endeavor).

In Section 4.3 of the revised manuscript, we discuss the impact of the use of fixed ocean volume and the restoring term for carbon isotopes on our results. Additionally, we discuss potential improvements in the experimental design based on the influence of forcings, considering the works by Snoll et al. (in review, 2023) and Bouttes et al. (2023). Both studies highlight that the choice of forcings, specifically freshwater fluxes and interactive bathymetry, has major impact on the time-series of the AMOC and associated climate changes. Nevertheless, our experimental design has the advantage of examining climate responses associated with abrupt climate changes (such as the Bølling–Allerød and Younger Dryas transitions) with minimal adjustment to freshwater. We acknowledge that there is uncertainty in the choice of climate forcing, and we recognize that improving the experimental design (such as incorporating interactive bathymetry and associated ocean volume changes) might influence the climate fields or the oceanic carbon content. We plan to address these factors in our future

studies.

5. Tense: I think that the past (e.g. L3, L8, L9··· and throughout the manuscript) and past perfect (e.g. L20, L22···) tense tend to make statements less effective than present tense.

We have edited and proofread the English text, carefully considering the impact of sentence structure in different tenses. We understand that it is customary to use the past tense when referring to our "study" because the reference is to work that was undertaken as part of the research effort (e.g., in the account of the study in the abstract and in the description of the research method in the Methods section). Descriptions regarding the content of the "paper," i.e., discussion of the results derived from the research effort of study, have been revised to the present tense.

**Specific comments:**

L5 and L7: "increased", "decreasing trend". By how much? More frequent quantification would be welcomed.

Thank you for your feedback. We have revised the abstract by incorporating more quantitative information.

L12-13: "We found that···". This statement seems to be contradictory with Fig. 2, unless "after the onset of the BA" is specified.

We have revised the abstract to address this concern. The text has been amended to mention changes in the AMOC after the onset of the Bølling–Allerød period.

L17-18: "smaller atmospheric $pCO_2$ changes than ice core data". Are you referring to changes during HS1 or during the whole deglaciation?

This statement refers to changes during Heinrich Stadial 1. We have restructured the abstract to make this aspect clearer.

L26-27: the "which is" proposition interrupts the "from ⋯ to⋯" statement, giving a jerky rhythm to the sentence.

We have revised the sentence to ensure smoother flow and improved clarity.

L31: An example of where a transition (e.g. "To decipher the reasons behind those changes,⋯") would be welcomed. Same for L34 (e.g. "In particular,⋯", with no line jump).

Thank you for highlighting the need for smoother transitions. We have incorporated these suggestions into the revised manuscript.

L37: "Therefore" provides for an incomplete argument for different $\delta^{13}$C in water masses, for only the impact of fractionation during photosynthesis is described before – not including the impact of ventilation on the isotopic signal.

We have added the following statement to the revised manuscript.

*"Anomalies in the stable carbon isotope signature ($\delta^{13}$C) produced by the biological carbon pump spreads globally following the deep ocean circulation."*

L54: Are you using the plural form to designate both the soft tissue and the carbonate pumps combined? According to Kohfeld and Rigwell (2009), strictly speaking, the biological pump (singular) encompasses both.

Thank you for this feedback. We have revised the manuscript to use the singular form *"biological pump"* in accordance with Kohfeld and Ridgwell (2009).

L55-59: I find the explanation of the limitations of the use of steady-state differences (and therefore, of the asset of using transient simulations) to be incomplete and therefore only partially convincing. Could you elaborate?

The revised manuscript includes information on the benefits of transient climate modeling over steady-state modeling, thereby underscoring the motivation behind our study.

*"Transient climate modeling has distinct advantages: it avoids unrealistic equilibrium assumptions and includes climate responses to internal variability or abrupt changes. It also facilitates direct comparisons between models and proxies, allowing us to identify time leads or lags in the process with respect to forcing."*

Steady-state differences tend to overestimate the impacts of processes operating on time scales of thousands to tens of thousands of years. For example, in our study (Kobayashi et al., 2021, *Sci. Adv.*; K21), we performed numerical integrations over hundreds of thousands of years to explore the steady-state response of carbonate sediments. However, because actual glacial–interglacial cycles and their transitions occur on shorter time scales, this approach might overestimate the realistic response. To better understand realistic processes, it is critical to examine the carbon cycle response that occurs within the transitions of climate change.

L59: The change of tense ("will improve" to "have been conducted") makes it confusing for the reader to understand what exactly is the knowledge gap, and what is new in this study with respect to previous studies.

Thank you for highlighting this issue. We have rephrased the wording to clarify the knowledge gap and to express the novelty of our study in comparison to previous research.

Previous studies using Earth Systems Models of Intermediate Complexity

(EMIC) have focused on atmospheric $CO_2$ fluctuations during glacial–interglacial cycles, consistent with ice core records. However, this approach has limitations such as overreliance on parameterized processes and poor comparison with observed sediment core records. Our research aims to fill this gap through numerical experiments on the ocean carbon cycle during the last deglaciation. We performed detailed model–data comparisons of carbon isotope ratios with recently compiled sediment core records to validate the simulated ocean carbon cycle changes, and to discuss possible biases and missing or underestimated processes in the model.

L66 and L71: the lack of transition with "other related studies", "several related studies" makes it difficult to follow the reasoning and understand where this paragraph is going. I would recommend connecting ideas rather than juxtaposing them.

We reorganized the paragraphs to create smoother transitions between ideas. The connections between *"other related studies"* and *"several related studies"* are now more clearly established, thereby improving the flow of the argument within the paragraph.

L71-75: this sentence contains many processes and seem therefore very long. Could it be divided into two? As for the references L75-76, it is unclear whether they refer to either one, or all, of the mentioned processes.

Following the advice, we have revised the sentence and split it into two for improved readability. Changes related to this issue are also reflected in our response to the previous feedback. The referenced studies broadly suggest that Southern Ocean processes are important regarding the rise in atmospheric $pCO_2$ during the last deglaciation. We have rewritten the relevant text in a more general sense and we have cited the references accordingly.

L78-80: it seems unclear to me (1) what were the conclusions of those

previous studies, (2) what is their limitation you are mentioning, and therefore (3) the novelty brought by your study. Could you elaborate?

The conclusions that can be drawn from the previous studies are summarized in the following.

(1) EMIC studies have conducted numerical experiments focusing on the changes in atmospheric $CO_2$ resulting from variations in oceanic and terrestrial carbon cycling during glacial–interglacial cycles or focusing on Heinrich Stadial 1. Some studies have successfully explained the amplitude of atmospheric $CO_2$ aligned with ice core records.

However, these studies are limited in respect to the following point.

(2) Although such results are an important advance, a large proportion of the amplitude is explained by the parameterized processes. Factors controlling temperature-dependent decomposition of organic matter, iron fertilization, $CO_2$ from hydrothermal venting, or gradual changes in meltwater and westerly wind strength are imposed. Furthermore, although these models reproduce changes in atmospheric $CO_2$ consistent with ice core records, comparisons of model output with observational sediment core records remain inadequate. These comparisons typically focus on one-dimensional vertical distributions of basin averages and lack detailed comparisons across different time slices of deglaciation.

In our study, we aimed to address this limitation.

(3) In this study, we performed numerical experiments on the ocean carbon cycle under deglacial climate changes derived from a climate model. We present model–data comparisons of carbon isotope ratios with recently compiled sediment core records for different time periods. We validate the simulated ocean carbon cycle changes and discuss possible biases and missing or underestimated processes in the model by comparing simulated

carbon isotope ratios with sediment core data.

Discussion of both the limitations of previous studies and the developments of this study is presented in the Introduction section of the revised manuscript.

L90: "boundary conditions" is a rather technical jargon which, depending on the model setup, can designate various things. I would prefer using "was forced with" instead.

Following the advice, the revised manuscript replaces the phrase *"boundary conditions"* with *"is forced with"* in several instances to improve the clarity of the text.

Sect. 2.2: A little surprisingly, the PMIP4 protocol (Ivanovic et al., 2016) is not mentioned. It could be worth noting whether the AOGCM simulation by OA19 followed the PMIP4 protocol for deglacial simulations, and if not, where it differed.

Reviewer #1 also made a similar comment. Section 2.2 of the revised manuscript provides a description of the MIROC-AOGCM experimental setup, identifying the components that follow the PMIP protocol (Ivanovic et al., 2016) and any subsequent modifications.

L106: Without reading the quoted paper, these values and how they were chosen seem a little mysterious.

The following description regarding the iron cycle model used in this study has been added to the revised manuscript.

*"The iron solubility is derived from the ratio of wet and dry dust deposition and its solubility. Note, however, that there is some uncertainty associated with these parameters."*

L115: Please give out here a quantified value of the pCO$_2$ simulated in the 2021 paper. Same in L162. The authors could also consider adding a triangle for this value in Fig. 2k.

We have added the quantified values of atmospheric pCO$_2$ simulated by K21; the triangles in Fig. 2k indicate specific pCO$_2$ values calculated by K21.

L124: Ocean volume change would also induce changes in alkalinity and nutrients, not only dissolved matter concentration.

As you correctly noted, changes in ocean volume can also lead to changes in alkalinity and nutrient concentrations. We have modified the sentence in the revised manuscript as follows: *"Notably, the transient experiment does not account for temporal variations in ocean volume caused by ice sheet changes and associated changes in mean ocean concentration of biogeochemical tracers (i.e., nutrients, alkalinity, and DIC)."*

L142: "underestimates". As the simulated values are less negative than the reconstructed ones, I am unsure whether "underestimates (how low these values are)" or "overestimates" should be used. This English vocabulary should be checked.

We have revised the description to clarify the meaning.

*"Notably, however, the simulated $\Delta\Delta^{14}C$ values are less negative than the reconstruction of $\Delta\Delta^{14}C$ at 21 ka BP in the Atlantic below 3000 m and in the Pacific below 2000 m."*

L155: "approximately 80 ppm". The value here is lower than the one chosen L28. It is best to use consistent numbers.

Thank you for highlighting this point. Throughout the revised manuscript, we use the value of *"approximately 80 ppm"* to represent the deglacial

change in atmospheric $pCO_2$.

L156 and L161: "One possible explanation" / "Another possible explanation". This vocabulary may give the reader the impression that these reasons are mutually exclusive, which is not the case.

As you correctly identified, these processes are not mutually exclusive and both can contribute. We have revised this text to indicate that these processes are not exclusive.

This description refers to the trend of $\Delta\Delta^{14}C$ throughout the last deglaciation. The $\Delta\Delta^{14}C$ in the deep ocean tends to gradually approach zero from the markedly negative values during the LGM. Variations in the deep ocean circulation disrupt this prominent trend.

*"Seawater $\Delta\Delta^{14}C$ generally increases from the relatively low LGM values, but decreases during HS1 and the YD."*

L175-178: This sentence brings very little new elements with respect to L174-175 (except the numerical value of -200 permil).

In acknowledgment of your observation, we have summarized the second half of this paragraph to avoid repetition.

L180-181: The link between the AMOC and the deep Pacific ventilation is unclear.

In this study, the volume transport of AABW increases during periods of intensified AMOC, such as the Bølling–Allerød period and the Holocene, which reduces $\Delta\Delta^{14}C$ in the Southern Ocean. This change is relatively small in the Pacific Ocean, excluding the Southern Ocean. This characteristic of

ventilation change might depend on the model and the background climate.

L183: "do not provide clear indication of the intrusion of young water masses". Where is this evidence? Please quote a figure number.

Figures 3 and S4 show the evidence for the statement *"do not provide clear indication of the intrusion of young water masses"*. For clarity, we have quoted the figure numbers.

L184-185: "whereas the model experiment does not show such pronounce change [during HS1]". I find this statement to be slightly misleading, for I would expect that the absence of a large increase of simulated Δ14C to be related not to a model error during HS1, but to the very high values (wrt. reconstruction) inherited from the initial state (i.e. not enough carbon sequestration in the ocean at the LGM).

Your point is quite valid. The differences between the model and the data during the early deglaciation are strongly influenced by the state of the carbon cycle inherited from the LGM; consequently, we have revised the relevant sentences.

L188: "Δ14C- $CO_2$ (Fig. 2a)". Please explain this choice of plot.

The intention is to show changes in atmospheric $\Delta^{14}C–CO_2$ attributable both to changes in the ocean carbon cycle and to exchange between the atmosphere and the ocean.

L190-195: This paragraph feels disconnected from the previous descriptions. A transition would probably help integrate it explicitly in the reasoning.

Thank you for pointing out the need for better integration in the text. We have revised this paragraph to ensure a smoother transition that explicitly aligns with the preceding descriptions:

*"Regarding the latter point of insufficient carbon sequestration during the LGM, the triangles shown in Fig. 2 represent the results of the best LGM simulation (LGM_all) conducted by Kobayashi et al. (2021)."*

L200: Could we perhaps see a figure of biological production as well in Supplementary?

We show a figure illustrating the changes in biological production from the LGM in Supplementary Fig. S6.

L199-201: How is a reduced vertically gradient in response to a stronger AMOC related to the sensitivity of $\delta$13C to climate change?

In this model, the basin-scale distribution of $\delta^{13}$C changes completely during the first 500 years or so when the AMOC changes from a weak state to a strong state. Given the time scales of ocean circulation, this result is not that surprising. Although we have not looked closely at the results of other models, it is possible that this response might vary on somewhat different time scales owing to counteracting action of the biological pump and to changes in terrestrial vegetation.

L204-206: The reasoning seems incomplete. How should we interpret this different model-data agreement for the North Atlantic / Southern Ocean?

During Heinrich Stadial 1, $\delta^{13}$C decreases gradually in the upper 3000 m of the North Atlantic (Figs. 2h, 4, and S3). The reduction in $\delta^{13}$C can be attributed to several factors that include an increased contribution from southern-sourced deep water with low $\delta^{13}$C endmembers, accumulation of remineralized carbon with low $\delta^{13}$C owing to a weakened AMOC and reduced ventilation, and increase in the $\delta^{13}$C endmember of the North Atlantic Deep Water (NADW) (Gu et al., 2021). In this study, no obvious change in the endmember of the NADW was observed. The $\delta^{13}$C change is attributed to weakening of the ventilation in the North Atlantic and to

expansion of southern-sourced deep water.

In the deep Southern Ocean, $\delta^{13}$C is lowest during the LGM and gradually increases during Heinrich Stadial 1. Such changes were not observed in the model. K21 explain the low $\delta^{13}$C during the LGM by considering the enhanced Southern Ocean stratification and iron fertilization from glaciogenic dust, which lower $\delta^{13}$C in the deep water. The absence of these processes is one factor explaining the difference between the model and the data.

The different model–data agreement between the North Atlantic and the Southern Ocean raises questions about the mechanisms governing the response of these regions to climate change. Further investigation is needed to fully interpret the disparity in model–data agreement and to better understand the different influences shaping $\delta^{13}$C dynamics in these specific oceanic zones. We have made revisions in the text with this specific point in mind.

L207: "the AMOC is intensified". Add "and deepens".

We have modified the text following your suggestion.

Sect 3.3.2 and 3.3.3: Like in Sect 3.3.1, I would welcome here a quantification of the atmospheric pCO$_2$ changes occurring during the BA and YD.

Sections 3.3.2 and 3.3.3 present specific quantitative data regarding changes in atmospheric pCO$_2$ during the Bølling–Allerød and Younger Dryas periods. The calculated changes in atmospheric pCO$_2$ are as follows.

Heinrich Stadial 1 (18–15 ka BP): increase in atmospheric pCO$_2$ by 10.2 ppm; Bølling–Allerød period (15–13 ka BP): decrease in atmospheric pCO$_2$ by 7.0 ppm; Younger Dryas period (13–12 ka BP): increase in atmospheric pCO$_2$ by

6.8 ppm.

L260: "the AMOC resumes over time". This feels like an inaccurate description of the AMOC variations, as Fig. 1 rather shows a stabilization during the BA after an overshoot at the onset of the BA.

The description of the AMOC has been revised to better reflect the fact that the post-overshoot stabilization phase of the AMOC coincides with the observed net increase in global biological production in the global ocean.

L280: "after the BA transition are generally consistent". Starting out this paragraph like this is a bit surprising (in terms of chronology), as this statement does not acknowledge the large model-data gap before the BA transition.

The revised manuscript is written in chronological order. After highlighting the major differences in $\Delta\Delta^{14}C$ during Heinrich Stadial 1, we discuss the changes that occurred after the Bølling-Allerød transition.

L293: "a longer period". Please quantify the difference. The forcing in terms of meltwater fluxes also seem to differ between the two studies.

These differences in duration have been clarified. The duration of the minimum AMOC during the Younger Dryas period is approximately 1000 years in our model, whereas it is approximately 500 hundred years in the Bern3D model. Moreover, the strength of the AMOC during the Younger Dryas period is approximately 4 Sv in our model and approximately 6 Sv in the Bern3D model.

[Figure]

Maximum Volume Transport of the AMOC

The maximum meridional volume transport of the AMOC between 30° N and 90° N and below 500 m depth.

L297-300: Although this seems like conclusive remarks, I am not coming out of the paragraph with a clear idea of what the comparison to the Pöppelmeier et al. (2023) results actually brought to the table. Clarifying the transition (L286, L301) could help.

Thank you for highlighting this area of concern. To more clearly express the comparison with the results of Pöppelmeier et al. (2023), we have revised the transition of these sentences.

Additionally, the descriptions of these lines that were noted have been moved to the Conclusions section.

L303: "to the deep ocean". Do you mean deep Atlantic ocean? This mismatch seems larger in the Atlantic than Southern Ocean.

As you correctly identified, the difference is most noticeable in the North Atlantic Ocean, which is greatly affected by NADW inflow. This sentence was a repeat of the previous sentence, and it has been removed because it was redundant.

L312-313: Could you specify? As such, this is an underwhelming statement.

Thank you for pointing this out. The description has been changed to specifically explain the factors that cause $\delta^{13}C$ variation.

"For $\delta^{13}C$, both the model and data show a similar, depicting an increase in deep water $\delta^{13}C$ during the BA as in $\Delta\Delta^{14}C$. However, there is a discrepancy during the following YD period. The model indicates a decrease in deep water $\delta^{13}C$ during the YD, whereas this feature is absent in the reconstruction (Figs. 2g--j). There are several possible factors that could potentially influence this discrepancy. Variations in change signals and potential dating inaccuracies within individual sediment core data can result from smoothing effects such as bioturbation and coring artifacts. These complexities highlight the need for caution when comparing model simulations and sediment core records. Another important factor is the weakening of the simulated AMOC during the YD. The comparison of the model and sediment data for $\Delta\Delta^{14}C$ and $\delta^{13}C$ suggests that the weakening of the AMOC during the YD may be overly pronounced in the model (Figs. 3e and 4e). In addition, the calculated increase in export of biogenic organic matter in the Southern Ocean during the YD compared to the BA (Figs. S6f and g) contributes to the decrease in $\delta^{13}C$ in the deep ocean. This emphasizes the importance of accurately simulating nutrient and iron cycles, especially in iron-limited regions affected by changes in dust-derived iron supply. As the Southern Hemisphere warms and becomes more humid, the supply of iron from dust may decrease (Martin, 1990; Martínez-García et al., 2014). Reproducing changes in $\delta^{13}C$ is challenging due to the intricate interconnections between ocean circulation, biological processes, and atmosphere-ocean gas exchange. Understanding the discrepancies between the model and data in $\delta^{13}C$ changes requires future sensitivity experiments to clarify their respective contributions and provide a deeper understanding of these factors."

L321: "AMOC weakening" could be changed to "AMOC weak state", since the actual weakening occurs at the onset of these events.

We have corrected it as indicated. Atmospheric $pCO_2$ increases when the AMOC is in the weak state.

L323: Shouldn't the contribution of SST during YD and HS1 also be mentioned?

In response to your question, we have modified the wording to emphasize that the major factors of changes in atmospheric $pCO_2$ in response to the AMOC fluctuations are alkalinity and SST changes.

L325: Could we perhaps see a figure with the vertical gradients in Supplementary?

We have added figures showing the changes in surface and deep DIC and alkalinity in the Atlantic and Pacific oceans to the Supplementary Figures.

L327: "increased SST and reduced surface ocean alkalinity". During which part(s) of the deglaciation specifically? Or do you mean the whole deglaciation?

The changes in "increased SST and decreased surface ocean alkalinity" are specific to the intervals of Heinrich Stadial 1 and the Younger Dryas period during the deglaciation.

L329: The absence of transition is more notable as we go back in time (reversed chronology here). L332: "In contrast" seems like the wrong link word here.

Thank you for bringing this to our attention. We have revised the text of the Discussion section, including improving transitions to create smoother flow.

L350-351: "AOGCM of MIROC", "EMIC of iLOVECLIM". The model classification is irrelevant here. The difference in model resolution could be mentioned if the authors would like to propose it as a potential cause for the observed model difference (although difference in forcings could also play a role).

As you correctly highlighted, the paper compares two coupled climate models with different responses to freshwater input and therefore the classification of the models is not essential. The description has been modified accordingly in the revised manuscript.

L420: Add something like "at the BA and YD transition respectively"

We have added the words that you have suggested.

*"To understand the mechanisms of glacial--interglacial variability in the carbon cycle, this study examines the transient response of the ocean carbon cycle to climate change, including the remarkable strengthening and weakening of the AMOC at the BA and YD transition respectively."*

L422: "relatively modest". Please quantify again.

We have added information regarding quantitative changes in atmospheric $pCO_2$.

Fig. 1: (a) Please specify in the legend what the metric used for the AMOC strength (max at 26°N?). Also, you could consider adding triangles for the Kobayashi et al. (2021) values as in Fig. 2.

The legend of the revised manuscript indicates that the metric used for "the AMOC strength" is defined as the maximum meridional volume transport between 30°N and 90°N at depths below 500 m. Additionally, we have included triangles to represent the atmospheric $pCO_2$ calculated in K21.

Fig. 2: The "PI_sed" and "LGM_all" simulations could be (briefly) described in the legend so that the reader knows in which way they differ from the transient run initial state without having to look for the simulation description in the 2021 paper.

We have added the experimental information of PI_sed and LGM_all to the legend of Fig. 2.

Fig. 3: Please specify which ocean basin is where on the plots (Atl = left, Pac = right).

We have added "ATL" and "PAC" to indicate the Atlantic Basin on the left and the Pacific Basin on the right side of the plots.

Fig. 4: The colorbar seems to saturate for very negative $\delta^{13}$C values in marine core data. Also, the contour interval is too dark and narrow to see to subsurface values. A few adjustments (and larger plots) could improve visibility.

We have modified the range of the contour lines, limiting them to exclude contours at data values below the minimum value or above the maximum value. This change improves the visibility of the figure.

Fig. 3 and 4: Please consider calculating the RMSE for both proxies and all periods.

We have calculated the correlation coefficient and RMSE for both proxies across all periods and added the information to Figs. 3 and 4.

Fig. 5: The thin gray line is not very visible on all panels. I would suggest finding an adjustment (e.g. lowering opacity for superimposed curves) to improve visibility.

To improve visibility, we have adjusted the image by changing the opacity of the overlaid lines.

**Technical comments:**

L2: "the effects" on what? Please specify.

We have revised the entire abstract and changed this wording.

L6: Introduce "(BA)" abbreviation here.

We have introduced the abbreviation "(BA)" at the point at which it is first used.

L10: "Meanwhile" or "in the meantime" L26: "has transitioned" → "transitioned"

We have revised this wording.

L43: typo "infer"

We have revised this wording.

L52: "those" → "the"

We have revised this wording.

L119: "ocean biogeochemical cycle" → add "carbon"

We have revised this wording.

L230: "of" → "the"

We have revised this wording.

L234: "those" → "the"

We have revised this wording.

L292: "almost completely" → drastically (or synonyms)

We have revised this wording.

L317: Add "⋯ to several factors. First, ⋯"; L319: "Moreover" → "Second"

We have added these transitional words.

L359: "are synchronized"

We have revised this wording.

L390: typo "lowering"

We have revised this wording.

L397: plural "developments"

We have revised this wording.

L436: "what" → "which"

We have revised this wording.

Most figures, especially Fig. 3 and 4, would gain to be larger.

The figures have been revised and enlarged to improve visibility.